# STAY HUNGRY, KEEP LEARNING: SUSTAINABLE PLASTICITY FOR DEEP REINFORCEMENT LEARNING

## ABSTRACT

The integration of Deep Neural Networks (DNNs) in Reinforcement Learning (RL) systems has led to remarkable progress in solving complex tasks but also introduced challenges like primacy bias and dead neurons. Primacy bias skews learning towards early experiences, while dead neurons diminish the network's capacity to acquire new knowledge. Traditional reset mechanisms aimed at addressing these issues often involve maintaining large replay buffers to train new networks or selectively resetting subsets of neurons. However, These approaches either incur substantial computational costs or fail to effectively reset the entire network, resulting in underutilization of network plasticity and reduced learning efficiency. In this work, we introduce the novel concept of neuron regeneration, which combines reset mechanisms with knowledge recovery techniques. We also propose a new framework called Sustainable Backup Propagation (SBP) that effectively maintains plasticity in neural networks through this neuron regeneration process. The SBP framework achieves whole network neuron regeneration through two key procedures: cycle reset and inner distillation. Cycle reset involves a scheduled renewal of neurons, while inner distillation functions as a knowledge recovery mechanism at the neuron level. To validate our framework, we integrate SBP with Proximal Policy Optimization (PPO) and propose a novel distillation function for inner distillation. This integration results in Plastic PPO (P3O), a new algorithm that enables efficient cyclic regeneration of all neurons in the actor network. This approach facilitates neuron regeneration while maintaining policy plasticity and sample efficiency. Extensive experiments demonstrate that, with proper neuron regeneration methods, the SBP framework can effectively maintain plasticity and improve sample efficiency in reinforcement learning tasks.

## 1 INTRODUCTION

Deep reinforcement learning has advanced significantly through the integration of deep neural networks, resulting in notable achievements across various domains(Singh et al., 2022; Arulkumaran et al., 2017; Yu et al., 2021). Despite these advancements, a critical issue that has emerged is the loss of plasticity, as detailed in (Lyle et al., 2023; Abbas et al., 2023). This refers to the diminishing ability of a network to learn and adapt over time. As network neurons become saturated, they "become full", losing the capacity to incorporate new information effectively. This reduction in plasticity primarily affects the neurons in the network, leading to decreased effectiveness and eventually causing neurons to become dead(Lu et al., 2019; Shin & Karniadakis, 2020) or dormant(Sokar et al., 2023). Additionally, the problem of overfitting in deep learning, known as primacy bias (Nikishin et al., 2022), further causes this loss of plasticity. Consequently, there is an urgent imperative to develop mechanisms for the repair or revitalization of neurons affected by primacy bias or those that have lapsed into dormancy, with the objective of reawakening their "hunger" for novel information.

Reset mechanisms have been proven to be effective measures for addressing the loss of plasticity in neural networks. However, existing reset approaches have demonstrated various limitations. Early studies (Nikishin et al., 2022; D'Oro et al., 2022; Kim et al., 2024) proposed resetting either the final layer or all neurons to revitalize learning capabilities. However, these methods often led to a performance-resource trade-off, requiring additional training to recover lost performance. More targeted approaches, such as CBP (Dohare et al., 2021) and ReDo (Sokar et al., 2023), focused on selectively resetting non-contributing neurons. While this strategy reduced information loss,

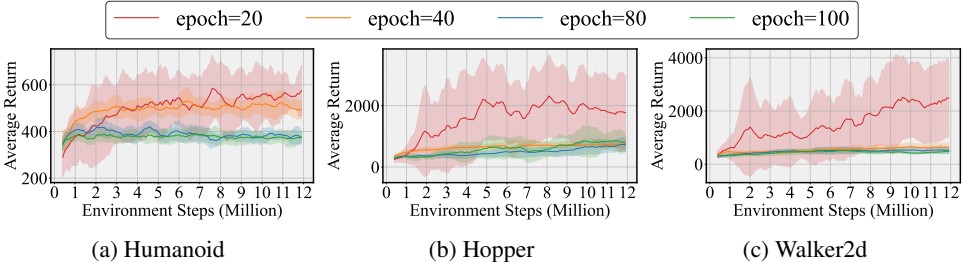

(a) Humanoid  (b) Hopper  (c) Walker2d

Figure 1: Performance of PPO across varying numbers of training epochs per batch. Increasing training steps impedes performance improvement.

it only partially restored plasticity and achieved limited performance improvements. A common challenge across these methods is that neurons reaching critical importance become either irresetable or damaging when reset, rendering the reset strategy ineffective and potentially limiting the network's overall potential. This underscores the need for a more sophisticated approach that can enhance network plasticity without compromising performance. Inspired by regenerative processes in human cells (Carlson, 2011), we propose the concept of *neuron regeneration*, which aims to recover plasticity while preserving crucial knowledge within the network.

*Neuron regeneration* refers to the periodic renewal of neurons to maintain network plasticity, drawing inspiration from biological cellular regeneration. This process enables neural networks to sustain adaptability and long-term learning capabilities without performance degradation. The key innovation lies in its ability to effectively recover plasticity while preserving network performance, thereby achieving sustainable plasticity. To implement *neuron regeneration*, we designed the Sustainable Backup Propagation (SBP) approach, which integrates reset and distillation mechanisms into traditional backpropagation(Hecht-Nielsen, 1992). Inspired by natural cellular regeneration cycles(Sender & Milo, 2021), SBP employs a cyclic reset strategy to mitigate neuron plasticity degradation and primacy bias. The *inner distillation* process facilitates knowledge transfer from reset neurons to others, ensuring effective regeneration without performance loss. This approach maintains the network's capacity to absorb new information, perpetuating its learning process and maximizing its potential for continuous growth. By balancing plasticity recovery with performance preservation, SBP enables neural networks to consistently maintain learning efficiency while enhancing plasticity, thus fully leveraging the network's learning capabilities.

As shown in Figure 1, Proximal Policy Optimization (PPO) (Schulman et al., 2017) suffers from primacy bias, with performance degrading as training epochs increase. This indicates a loss of plasticity over time, highlighting the need for sustainable plasticity. To address this, we integrated our Sustainable Backup Propagation (SBP) approach with PPO, introducing the concept of neuron regeneration. We developed a novel $\alpha$-weighted Double KL divergence ($\alpha$-DKL) loss function to preserve useful knowledge while filtering out harmful information. This integration resulted in the Plastic PPO (P3O) algorithm, which implements neuron regeneration within the actor network to maintain learning efficiency and achieve sustainable plasticity. We evaluated P3O across various environments, including MuJoCo (Todorov et al., 2012), DeepMind Control Suite (Tassa et al., 2018), and a MuJoCo variant called Cycle Friction. Results demonstrate significant performance improvements. These findings validate the effectiveness of our neuron regeneration mechanism in recovering plasticity and enhancing learning capabilities in reinforcement learning tasks. Our contributions in this work can be summarized as follows:

- **Neuron Regeneration:** We introduce the concept of *neuron regeneration*, a novel approach inspired by biological processes to maintain and recover plasticity in neural networks, enabling continuous adaptation without performance degradation.

- **Sustainable Backup Propagation (SBP):** We propose the *SBP* framework, which implements neuron regeneration through cyclic reset and inner distillation. SBP effectively addresses issues such as dead neurons and primacy bias, ensuring sustainable plasticity throughout the network's lifecycle.

- **Plastic PPO (P3O):** We introduce P3O, an enhanced version of PPO that integrates SBP and a novel $\alpha$-weighted Double KL divergence ($\alpha$-DKL) loss function. P3O overcomes

the primacy bias problem in standard PPO, maintaining plasticity and improved sample efficiency across various reinforcement learning tasks.

## 2 PRELIMINARIES AND RELATED WORK

### 2.1 ON-POLICY REINFORCEMENT LEARNING

Reinforcement learning is formalized as a Markov decision process (MDP) (Puterman, 2014). An MDP consists of a tuple $\langle S, A, R, P, \gamma \rangle$, where $S$ denotes the set of states, $A$ the set of actions, $R : X \times S \to \mathbb{R}$ a reward function, $P : S \times A \to P($ a possibly stochastic transition probability function, $\gamma \in [0, 1)$ the discount factor. In reinforcement learning, The goal is to seek an optimal policy $\pi^* : S \to P(A)$ which maximizes the expected accumulated returns with discounted.

In on-policy reinforcement learning, the Proximal Policy Optimization (PPO) algorithm (Schulman et al., 2017) is utilized to update policies during interaction with the environment. The core objective function of PPO, denoted as $\mathcal{L}_{\text{clip}}(\theta)$, includes a clipping operation that sets gradients to zero when the probability ratio $r_t(\theta)$ falls outside $[1 - \epsilon, 1 + \epsilon]$. This prevents the policy from learning from advantages that would push it further outside the trust region, thereby enforcing trust region constraints in a computationally efficient manner. This operation ensures that the policy does not deviate excessively from the previous policy. The objective function is expressed as:

$$\mathcal{L}_{\text{clip}}(\theta) = \mathbb{E}_t \left[ \min \left( r_t(\theta) \cdot \hat{A}_t, \text{clip} \left( r_t(\theta), 1 - \epsilon, 1 + \epsilon \right) \cdot \hat{A}_t \right) \right] \tag{1}$$

Here, $r_t(\theta) = \frac{\pi_\theta(a_t|s_t)}{\pi_{\theta_{\text{old}}}(a_t|s_t)}$ represents the probability ratio between new and old policies, and $\hat{A}_t$ is the estimated advantage function. The advantage function measures the value of the current policy relative to a baseline policy, calculated from trajectory data acquired during interactions with the environment. In on-policy training environments with dynamic data, where both input data and target values are nonstationary, network plasticity may be compromised, leading to suboptimal performance.

### 2.2 PLASTICITY IN REINFORCEMENT LEARNING

Plasticity loss in neural networks, the gradual decline in learning capacity over time, poses a critical challenge in deep learning by limiting adaptation to new tasks. This issue has prompted extensive research(Lyle et al., 2023; Abbas et al., 2023; Nikishin et al., 2022; Nauman et al., 2024; Dohare et al., 2024) into methods for preserving neural network plasticity, with two main categories of approaches emerging to address this problem. The first involves various training techniques such as regularization, adjustments in activation functions, weight decay, and normalization strategies (Kumar et al., 2023; Delfosse et al., 2021; Lee et al., 2024; Lyle et al., 2024). These methods have proven somewhat effective at alleviating the problem by reducing overfitting and preventing parameters from becoming overly large, thereby delaying the onset of plasticity loss.

The second category of approaches involves resetting the network (Nikishin et al., 2022; Schwarzer et al., 2023; Nikishin et al., 2024) to recover the plasticity. This reset-based methodology addresses both neuron dormancy and primacy bias. Typically implemented by reinitializing weights of specific layers or the entire network, resets have been shown to effectively scale replay ratios, contributing to performance improvements (Kim et al., 2024; Xu et al., 2023). However, while resets can revive learning capabilities, they might lead to temporary performance degradation and require additional training to restore previously learned information. To mitigate these drawbacks, methods like CBP (Dohare et al., 2021) and ReDo (Sokar et al., 2023) selectively reinitialize neurons deemed less useful based on certain metrics, minimizing the impact on overall performance. This approach highlights the delicate balance between recovering plasticity and maintaining network efficiency. However, there remains a need for more sophisticated methods that can effectively regenerate neurons while preserving learned knowledge, which is the focus of our proposed approach.

The importance of addressing plasticity loss is further underscored by recent studies. These have demonstrated that enhancing neural plasticity can lead to significant improvements in sample efficiency (D'Oro et al., 2022; Schwarzer et al., 2023; Lee et al., 2024; Ma et al., 2023; Nauman et al., 2024), a particularly attractive property for reinforcement learning tasks where data efficiency is crucial.

## 2.3 Policy Distillation in Reinforcement Learning

Policy distillation is a technique that involves the extraction and transfer of knowledge or policy from a reinforcement learning agent to a smaller network, thereby improving efficiency (Rusu et al., 2015). Prior research (Igl et al., 2020; Lyle et al., 2022) has highlighted its utility in addressing potential generalization loss in deep reinforcement learning agents due to nonstationarity and overfitting, respectively. This suggests that policy distillation serves two main functions: transferring knowledge and enhancing generalization, aligning well with the goals of plasticity recovery such as preserving knowledge and mitigating primacy bias. A critical aspect of effectively implementing policy distillation is quantifying the quality of knowledge transfer, which necessitates appropriate divergence measures. A study by (Martins et al., 2021) discusses two types of Kullback-Leibler (KL) divergence measures: Forward KL (FKL) and Reverse KL (RKL). The Forward KL divergence, $D_{\mathrm{KL}}^{\rightarrow}$, weights the state space according to the teacher's policy, prioritizing learning in states where the teacher's policy is more probable. Conversely, the Reverse KL divergence, $D_{\mathrm{KL}}^{\leftarrow}$, weights according to the student's policy, promoting exploration and robustness but risking neglect of some teacher-favored behaviors. Their mathematical expressions are:

$$D_{\mathrm{KL}}^{\rightarrow}(\pi_1 \parallel \pi_2) = \sum_{s \in \mathcal{S}} \pi_1(s) \log\left(\frac{\pi_1(s)}{\pi_2(s)}\right) \tag{2}$$

$$D_{\mathrm{KL}}^{\leftarrow}(\pi_2 \parallel \pi_1) = \sum_{s \in \mathcal{S}} \pi_2(s) \log\left(\frac{\pi_2(s)}{\pi_1(s)}\right) \tag{3}$$

Recognizing the complementary strengths of these measures, our work proposes an integrated FKL-RKL approach, aiming to enhance knowledge transfer for plasticity recovery.

## 3 Neuron Regeneration

Maximizing the utilization of a neural network's capacity is a primary goal in training, yet several challenges impede this objective. Plasticity in neural networks, often viewed as a consumable resource, diminishes as the network learns and integrates knowledge. This depletion can result from suboptimal configurations such as inappropriate activation functions (Abbas et al., 2023), poor data quality (Lee et al., 2024), or inherent limitations of backpropagation (Dohare et al., 2024), leading to biased or irrelevant information acquisition and suboptimal network performance.

While neuron reset techniques can restore plasticity, they risk performance degradation if not carefully implemented(Nauman et al., 2024). Research by D'Oro et al. (2022) suggests that resetting all neurons, including those affected by primacy bias, is necessary to maximize network capacity. Building upon the idea of neuron reset while addressing its limitations, we introduce a new concept: neuron regeneration.

> **Definition 3.1: Neuron Regeneration**
>
> Given a neural network parameterized by $\theta$, let $NR$ denote a regeneration operation that resets arbitrary neurons to their initial plastic states, resulting in a new parameter set $\theta'$. Neuron regeneration maintains two key properties:
> 1. Neuron-level Plasticity Recovery: $\text{Plasticity}(NR(\theta')) > \text{Plasticity}(\theta)$
> 2. Performance Guarantee: $P(NR(\theta')) \geq P(\theta)$
> where Plasticity denotes a measure of the network's adaptability and $P$ represents the network's performance.

Neuron regeneration aims to achieve sustainable plasticity without performance degradation, enabling long-term learning and exploration of the network's potential. This approach addresses the limitations of traditional training methods and reset techniques, potentially maximizing the utilization of the entire network's capacity. By implementing neuron regeneration, we propose a framework for maintaining learning ability and efficiency, pushing the boundaries of neural network capabilities beyond current limitations.

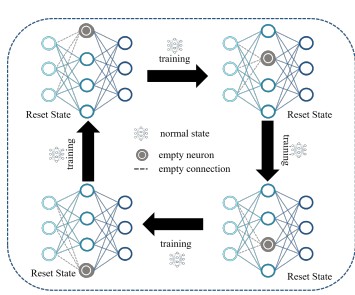 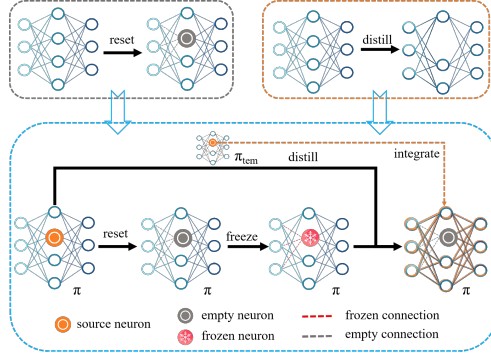

Figure 2: Left: Cycle Reset. Right: Inner Distillation. The right figure illustrates our method combining neuron reset and knowledge distillation. This approach restores neuron plasticity and transfers knowledge, enhancing overall network plasticity while maintaining performance.

## 4 METHODOLOGY

To maximize the utilization of neural network capabilities, we propose the Sustainable Backpropagation (SBP) method. SBP achieves sustainable plasticity through a novel neuron regeneration mechanism, which combines two key components: Cycle Reset and Inner Distillation.

### 4.1 SUSTAINABLE BACKPROPAGATION

Our neuron regeneration mechanism consists of two complementary processes: Reset and Distillation. The Reset process discards outdated knowledge and rejuvenates inactive neurons by resetting their parameters to initial states. Complementing this, the Distillation process preserves valuable knowledge by transferring information from the pre-reset state to the post-reset network. This combination allows neurons to restore plasticity while retaining crucial learned information.

To implement network-wide neuron regeneration, we introduce the Cycle Reset mechanism, governed by two key parameters: Reset frequency $F$ and Reset ratio $p$. Every $F$ training steps, $p\%$ of neurons in each layer undergo reset and regeneration. This cyclical process, illustrated in Figure 2, continuously refreshes neurons across all layers throughout training.

Inner Distillation completes the neuron regeneration process initiated by Reset. As shown in Figure 2, before resetting, the current policy $\pi_\theta$ is copied to a temporary policy $\pi_{\text{tem}}$. Selected neurons then undergo reset, clearing outdated information. Subsequently, knowledge from $\pi_{\text{tem}}$ is selectively distilled back to the reset policy $\pi_{\theta'}$ at the neuron level, with reset neurons temporarily frozen to preserve their renewed plasticity. This neuron-specific distillation process complements the Reset operation, jointly achieving neuron regeneration. The integration of Cycle Reset and Inner Distillation results in a cyclic neuron regeneration process. This approach ensures that all neurons in the network undergo periodic regeneration, maintaining network-wide plasticity while preserving valuable knowledge. Through this cyclic neuron regeneration, we achieve sustainable plasticity, enabling the neural network to continually adapt to new information without compromising its learned capabilities. This process maximizes the utilization of the neural network's capacity, effectively balancing ongoing learning with performance preservation.

The SBP algorithm, which incorporates reset operations and distillation, is outlined in Algorithm 1.

### 4.2 DOUBLE KL DIVERGENCE

Our analysis of Figure1 revealed that the PPO algorithm faces challenges with plasticity loss. To address this issue, we propose integrating the SBP framework into PPO, aiming to enhance its capabilities. The Inner Distillation process presents a complex scenario where neurons may contain either valuable knowledge or irrelevant information. To maximize generalization and plasticity, we must carefully control this distillation process. In the context of reinforcement learning, we leverage the KL divergence as our distillation function, building upon previous research(Martins et al., 2021)

that demonstrated the effectiveness of both Forward KL (FKL) and Reverse KL (RKL) in different aspects of knowledge transfer. Specifically, FKL has shown efficiency in transferring knowledge from a teacher policy to a student policy, while RKL is effective in preventing the infiltration of potentially harmful knowledge into the student model. To capitalize on the strengths of both approaches, we introduce a combined method. We also incorporate a parameter $\alpha$ to adapt the loss function to various distillation scenarios. This results in our proposed $\alpha$-weighted Double KL divergence (DKL), expressed as:

$$\mathcal{L}(\theta) = \min_{\theta} \alpha \cdot D_{\text{KL}}^{\rightarrow}(\pi_{tem} \parallel \pi_{\theta'}) + (1 - \alpha) \cdot D_{\text{KL}}^{\leftarrow}(\pi_{\theta'} \parallel \pi_{tem}) \tag{4}$$

- $\pi_{\theta}$ represents the primary policy, $\pi_{\text{temp}}$ denotes a temporary policy typically used before a reset, and $\pi_{\theta'}$ signifies the policy after the reset of certain neurons.

- $D_{\text{KL}}^{\rightarrow}(\pi_{\text{tem}} \parallel \pi_{\theta'})$ denotes the Forward KL divergence, which measures how well the policy $\pi_{\theta'}$ approximates the policy $\pi_{\text{tem}}$. This term is crucial for the effective transfer of essential knowledge from the $\pi_{\text{tem}}$ to the $\pi_{\theta'}$.

- $D_{\text{KL}}^{\leftarrow}(\pi_{\theta'} \parallel \pi_{\text{tem}})$ represents the Reverse KL divergence, which acts as a regularizer to prevent the $\pi_{\theta'}$ from adopting potentially harmful or irrelevant information from the policy $\pi_{\text{tem}}$.

- $\alpha \in [0, 1]$ is a tuning parameter that balances the contributions of the Forward and Reverse KL divergences to the overall loss function.

The $\alpha$-DKL approach offers a flexible and robust method for knowledge distillation, allowing us to balance the transfer of useful information with the prevention of harmful knowledge infiltration. By adjusting the $\alpha$ parameter, we can fine-tune the distillation process to suit different learning scenarios and optimize the trade-off between knowledge preservation and plasticity restoration. Leveraging $\alpha$-DKL, we propose Plastic PPO (P3O), an enhanced version of the PPO algorithm that integrates SBP and employs $\alpha$-DKL as its distillation loss function. The details of P3O are presented in Algorithm 2. This integration allows P3O to maintain sustainable plasticity throughout the learning process, potentially overcoming the limitations observed in standard PPO implementations.

## 5 EXPERIMENTS

### 5.1 EXPERIMENTAL SETUP

**Environment & Task** To evaluate our algorithm's performance, we employed a diverse set of tasks. These include standard benchmarks from MuJoCo (Todorov et al., 2012) and the state-based versions of DeepMind Control Suite (DMC) (Tassa et al., 2018). Additionally, we introduce the Cycle Friction Control task, an innovative variant of the MuJoCo environment inspired by the slip MuJoCo task (Dohare et al., 2024). Fig. 3 shows a task with a cyclically changing friction coefficient. It starts at 4, decreases by 1 every million steps to 1,

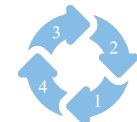

Figure 3: Cycle Friction.

then increases back to 4. This discrete evolution significantly increases environmental complexity, challenging the algorithm.

**Baseline** Throughout the entire experiment, we employed PPO as the base algorithm. In the reset experiment, we examined the impact of various reset strategies. **CBP** (Dohare et al., 2021) involves selecting neurons based on a utility function that considers both weight and activation values. **ReDo** (Sokar et al., 2023) selects neurons based on a score derived from their activation values. **Cycle** involves selecting neurons in a specific order.

Table 1: Performance comparison across MuJoCo environments, with results averaged across 5 different random seeds. Percentages show improvement over baseline PPO. H-stand: HumanoidStandup, Half: HalfCheetah.

|          | PPO     | PPO+CBP           | PPO+ReDo          | PPO+Cycle         | P3O               |
|----------|---------|-------------------|-------------------|-------------------|-------------------|
| Hopper   | 3,613   | 3,606 (-0.19%)    | **3828(5.95%)**   | 3280 (-9.22%)     | 3,735 (3%)        |
| H-stand  | 144,831 | 143,763(-0.74%)   | 151,254 (4.43%)   | 154323 (6.55%)    | **158,361 (9%)**  |
| Walker   | 5,878   | 5,028 (-14.46%)   | 5,594 (-4.83%)    | 5743 (-2.30%)     | **7,402 (9%)**    |
| Ant      | 3,514   | 4,860(38.30%)     | 3995 (13.69%)     | 3256 (-7.34%)     | **5,683 (62%)**   |
| Half     | 4,575   | 5,458 (19.30%)    | 4,962 (8.46%)     | 4843 (5.86%)      | **9,065 (98%)**   |
| Humanoid | 972     | 3,307 (240.23%)   | 2,573 (164.71%)   | 1578 (62.35%)     | **7,469 (669%)**  |

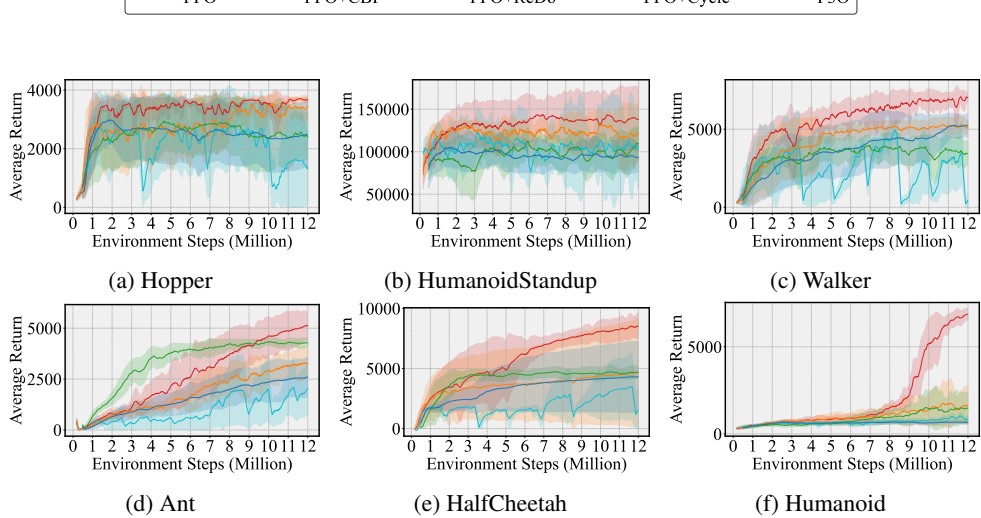

Figure 4: Performance of various reset strategies in MuJoCo environments.

## 5.2 Experimental Results

**MuJoCo** Our experimental results demonstrate the effectiveness of our P3O algorithm across several key dimensions. Table 1 shows that P3O consistently outperforms other algorithms across all MuJoCo environments, with particularly remarkable improvements in the Humanoid environment, where the performance boost reaches 669%. This substantial increase in maximum rewards indicates that our framework effectively enables neuronal regeneration, enhancing learning efficiency and more fully utilizing the neural network's capacity. The learning curves in Figure 4 further support this, demonstrating that P3O achieves the highest learning efficiency in most environments and maintains an upward trend for extended periods. This sustained improvement suggests that we have indeed achieved sustainable plasticity, continuously providing the neural network with the capacity for learning and adaptation. Moreover, our ablation study using a standalone cycle reset without distillation demonstrates that knowledge recovery plays a crucial role in this process, further validating the importance of our integrated approach in P3O.

As illustrated in Figure 5, we analyzed the average L1 norm of all neuron weights in the network during training. We observed that the original PPO algorithm tends to produce larger weights. In contrast, algorithms incorporating resets, including P3O, maintain weights in a more stable, lower range. This aligns with research(Dohare et al., 2024) indicating that excessively large weights are a symptom of reduced plasticity. Our findings, detailed in Table 4 in the appendix, show a correlation between reset frequency and weight magnitude: more frequent resets lead to smaller overall weights. P3O, with its moderate reset frequency, achieves a balanced weight distribution, positioned between the frequent resets of CBP and the limited resets of ReDo. This observation demonstrates that

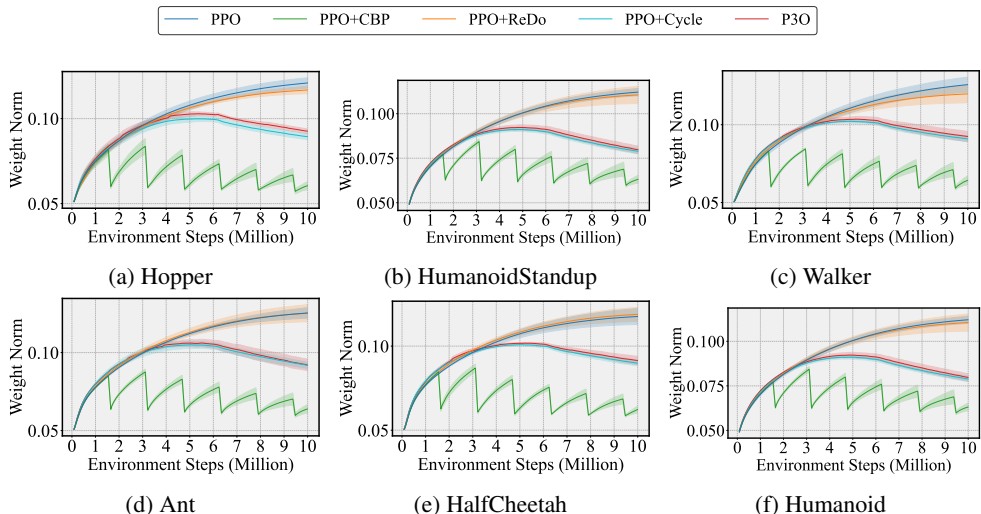

Figure 5: Weight Norm of actor network. Lower weight norm indicates greater plasticity.

resetting is indeed an effective measure for restoring plasticity, as it helps to maintain weights within a more optimal range for learning.

Crucially, as illustrated in Figure 6, P3O consistently maintains higher gradient norms compared to other tested algorithms across various MuJoCo environments. This is significant because learning occurs through gradients, and within a certain range, higher gradients indicate higher learning efficiency and plasticity. The phenomenon of vanishing gradients is often associated with a loss of plasticity. Our algorithm effectively reduces weight values while maintaining robust gradients, resulting in optimal performance. Other algorithms show less effective gradient maintenance, suggesting that knowledge retention is crucial for sustaining gradients.

The superior performance of P3O can be attributed to its appropriate reset frequency and effective knowledge retention measures. This combination enables efficient neuronal regeneration, balancing adaptation to new information with preservation of essential knowledge. By effectively navigating the plasticity-stability trade-off, P3O enhances sample efficiency in reinforcement learning tasks while maintaining learning stability. Moreover, P3O's success highlights the importance of neuronal regeneration and sustainable plasticity in neural networks. It demonstrates how continuous renewal of neuron, coupled with crucial information preservation, can achieve long-term adaptability without compromising performance or stability.

**Distillation Cost** Distillation is conducted in epochs using an online replay buffer, allowing us to calculate the cost of distillation by statistically analyzing its epochs. Table 2 presents these specific statistical results. However, indiscriminately increasing the number of epochs can exacerbate the primacy bias (Nikishin et al., 2022). This phenomenon is also observed in PPO, as demonstrated in Fig. 1. Thus, a strategic combination of distillation and reset mechanisms is crucial to optimize their benefits. Analysis of distillation epochs and their impact on P3O performance, as shown in Figure 7, reveals a clear trend: more training epochs lead to greater performance improvements. This suggests that distillation not only recovers past knowledge but also

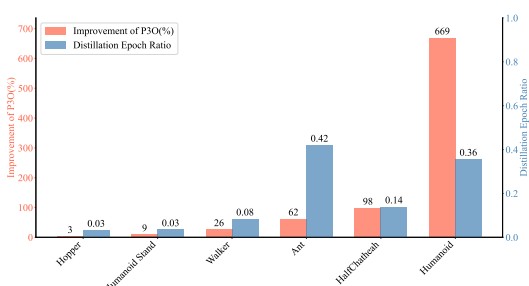

Figure 7: Correlation between distillation epochs and performance improvements.

improves sample efficiency. Crucially, the extra training epochs necessitated by distillation should be viewed as a key advantage of our algorithm, not a drawback. This strategic use of additional epochs significantly enhances sample efficiency, which is the primary reason our algorithm outperforms baseline algorithms such as CBP and ReDo. By leveraging these increased training steps, our frame-

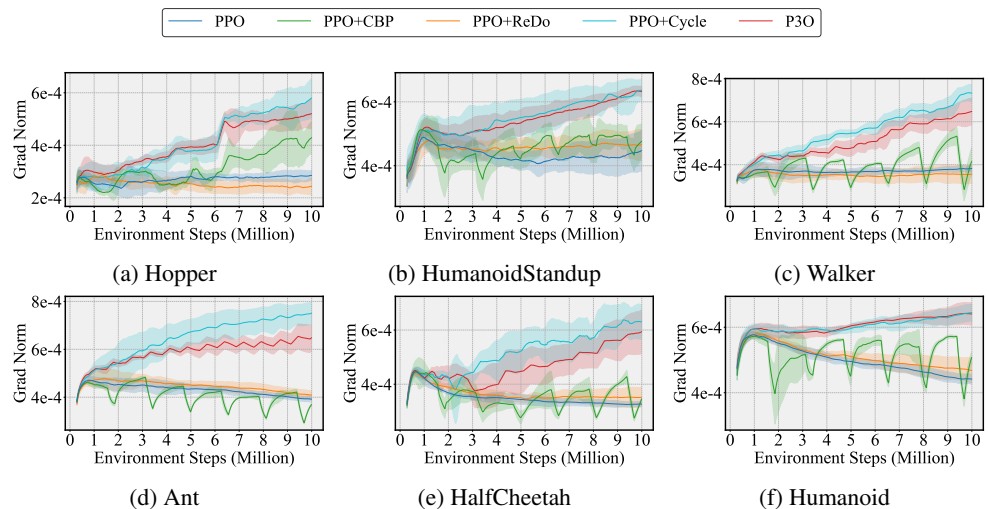

Figure 6: Grad Norm of actor network. Higher gradient norm indicates greater plasticity.

work aligns well with prior research on replay ratios in off-policy learning, while providing superior performance through improved sample utilization.

Table 2: Extra distillation epochs in P3O vs. PPO baseline (18,310 epochs) across environments.

|                 | Hopper | Humanoid Stand | Walker | Ant    | HalfCheetah | Humanoid |
|-----------------|--------|----------------|--------|--------|-------------|----------|
| Total Epochs    | 597.66 | 638.19         | 1536.80| 7698.40| 2548.40     | 6525.25  |
| Per-Reset Epochs| 2.39   | 2.55           | 6.14   | 30.79  | 10.19       | 26.10    |
| Epoch Ratio     | 0.0327 | 0.0349         | 0.0840 | 0.4204 | 0.1392      | 0.3564   |

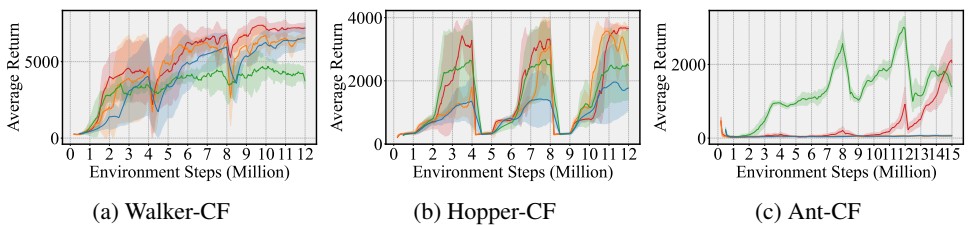

Figure 8: Performance of various reset strategies in Cycle Friction(CF) environments.

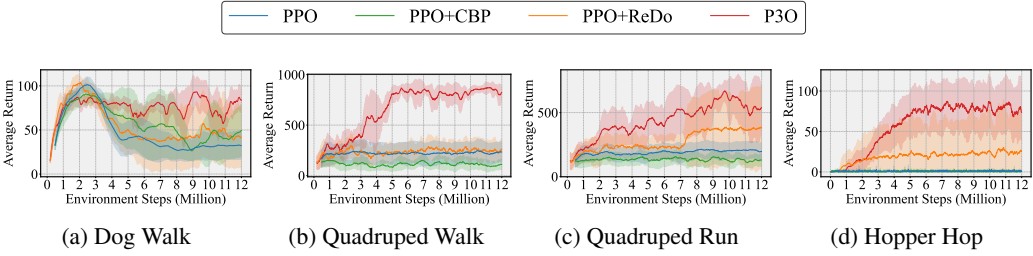

Figure 9: Performance of various reset strategies in DMC environments.

**DMC & Cycle Friction** Figures 8 and 9 reveal that in these complex benchmarks, while PPO struggles and other algorithms show only marginal improvements, our SBP approach achieves substantial progress. This superior performance in challenging environments demonstrates our algorithm's effectiveness and indicates that increased environmental complexity demands higher neural plasticity.

Our method's success underscores the importance of effective neuronal regeneration in complex tasks. These findings not only validate our approach but also highlight the need for further research into maximizing neural plasticity, especially in intricate learning environments.

The experimental outcomes observed across the Humanoid 4, Hopper Hop 8, and Cycle Friction Ant 9 environments demonstrate that it is often the constrained plasticity of neural networks, rather than the inherent limitations of the algorithms, that restricts the acquisition of valuable knowledge from data. This insight emphasizes the critical need for further research into the plasticity of neural networks, highlighting its pivotal role in advancing learning capabilities.

### 5.3 ABLATION OF **DISTILLATION** $\alpha$

The alpha parameter plays a role in controlling knowledge transfer during distillation, potentially influencing the effectiveness of neural regeneration. Our experiments, as illustrated in Figure 10, suggest that different alpha values can affect learning efficiency.

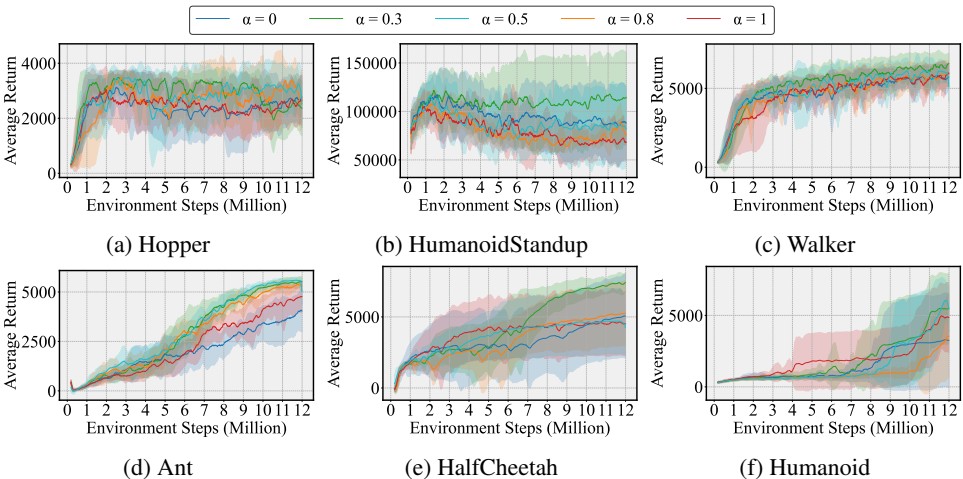

Figure 10: Performance of P3O with varying $\alpha$ values across Mujoco environments.

Our experiments show that an alpha of 0.3 generally yields better performance, suggesting that limiting forward-propagated information may be beneficial. This finding hints at the presence of primacy bias in current learning frameworks, potentially hindering ongoing performance improvements. The effectiveness of a lower alpha value indicates that the current learning paradigm may capture a limited amount of truly beneficial knowledge. This observation underscores the importance of resetting a significant portion of neurons to mitigate biased information learned early in training. Consequently, effective neuron regeneration becomes crucial to balance the preservation of valuable learned information with the network's ability to acquire new knowledge.

## 6 CONCLUSION

In this work, we introduced the Sustainable Backup Propagation(SBP) framework, centered on the concept of neuronal regeneration to achieve sustainable plasticity in neural networks. Our approach aims to maximize the utilization of neural networks by maintaining their long-term learning capacity. The core components of SBP, including a novel distillation mechanism and strategic reset mechanism, work in concert to preserve valuable knowledge while enabling continuous adaptation to new information. We implemented this framework in Plastic PPO(P3O) algorithm, which demonstrated significant performance improvements across various reinforcement learning tasks. Our findings highlight the potential of neuronal regeneration and sustainable plasticity as critical components in advancing deep learning paradigms. By addressing the plasticity-stability trade-off, SBP and P3O pave the way for more adaptive and efficient neural networks. Despite our significant progress, we acknowledge that in some environments, our approach did not achieve optimal performance. This suggests that neuronal regeneration requires further research to reach its full potential.

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

# A APPENDIX

## A.1 HYPERPARAMETER

- Python 3.8
- Pytorch 2.0.1 (Paszke et al., 2019)
- Gym 0.23.1 (Brockman et al., 2016)
- MuJoCo 2.3.7 (Todorov et al., 2012)
- mujoco-py 2.1.2.14

Our experiment is based on PPO and incorporates SBP, CBP, and ReDo variants. We use the hyperparameters described in Table 3 for all algorithms. It's important to note that we've slightly modified the ReDo mechanism: instead of using a threshold-based selection, each reset is based on evaluating the scores of the bottom one percent of neurons. To ensure statistical significance and reproducibility, all experiments were conducted with 5 different random seeds, and the results presented are the mean values with corresponding standard deviations across these runs.

Table 3: Algorithm Parameters

| Category | Hyperparameter | Value |
| --- | --- | --- |
| | Optimizer | Adam (Kingma & Ba, 2014) |
| | Learning rate (Actor & Critic) | 3e-4 |
| | Online replay buffer size | 8192 |
| PPO | Mini-batch size | 256 |
| | Discount factor | 0.99 |
| | Training steps | 1.25e7 / 2e7 |
| | Epochs per update | 10 |
| | Clip range | 0.2 |
| | Actor & Critic hidden dim | 256 |
| Architecture | Actor & Critic hidden layers | 3 |
| | Actor & Critic activation function | tanh |
| | Reset Rate | 0.01 |
| | Reset Frequency | 50000 Environment Step |
| SBP | Neuron Utility Type | Neuron lifetime |
| | Distillation $\alpha$ | 0.4 |
| | $\tau$ | 0.01 |
| | Reset Rate | 0.01 |
| CBP | Reset Frequency | 10000 Gradient Step |
| | Neuron Utility Type | Contribution |
| | Reset Rate | 0.01 |
| ReDo | Reset Frequency | 50000 Environment Step |
| | Neuron Utility Type | ReDo score |

## A.2 COMPARISON OF RESET STRATEGIES

In our study, we made deliberate choices in reset counting methods to align with each algorithm's characteristics while maintaining comparability. For CBP, we retained its original approach of using gradient steps for reset counting, preserving its algorithmic features. In contrast, for P3O, we opted to use environment interaction steps as the basis for reset counting. This decision was motivated by our focus on understanding how changes in input data affect neural plasticity. For consistency and to facilitate better comparison, we applied this same counting method based on environment interaction steps to ReDo as well. This approach allowed us to maintain the unique aspects of each algorithm while ensuring a meaningful comparative analysis across different reset strategies.

Table 4 presents a comparative analysis of neuron reset statistics for the CBP, P3O, and ReDo algorithms throughout their respective training processes. The data represents the average across six Mujoco environments. Our findings reveal distinct patterns in reset frequency and scope among these algorithms:

- CBP exhibits the highest reset frequency, followed by P3O, with ReDo having the least frequent resets.

- In terms of reset scope, both CBP and P3O can reset all neurons, while ReDo has limitations in this aspect.

These reset patterns align with the weight norm distributions observed in Figure 5. The data suggests an inverse relationship between reset frequency and weight magnitude: more frequent resets correspond to smaller neuronal weights. This observation unveils a simple yet significant principle: the more frequent and comprehensive the resets, the smaller the neuron weights tend to be. These findings highlight the importance of carefully considering reset strategies in the design and implementation of neural plasticity algorithms, as they can significantly affect the learning dynamics and ultimate performance of the models.

Table 4: Reset Statistics Comparison (768 Neurons)

|      | Total Resets | Average Resets | Reset Proportion (%) |
|------|--------------|----------------|----------------------|
| CBP  | 44208.6      | 57.6           | 100.0                |
| ReDo | 1800.0       | 2.3            | 22.3                 |
| SBP  | 2304.0       | 3.0            | 100.0                |

---

**Algorithm 1** Sustainable Backup Propagation (SBP)

---

Neural Network $f_\theta$, Temporary Model $f_{\text{tmp}}$, Reset Rate $\gamma$, Training Steps $T$, Reset Frequency $F$, Reset Index $p = 0$, Distillation Threshold $\tau$, Distillation Loss $d = $ None. **for** $t = 1$ *to* $T$ **do**

    Update Neural Network $f_\theta$ with standard backpropagation.

    **if** $t \mod F = 0$ **then**

        Copy the weights of Neural Network $f_\theta$ to Temporary Model $f_{\text{tmp}}$.

        **for** *each layer $L$ of the Network* **do**

            Let $l = $ neurons of layer $L$.; Reinitialize input weights of neuron $i$ in layer $L$: $i \in [p \cdot l : (p + \gamma) \cdot l]$.

        Freeze all the reset neurons in Neural Network $f_\theta$.

        **while** $d > \tau$ *or* $d = $ *None* **do**

            Update Neural Network $f_\theta$ using Temporary Model $f_{\text{tmp}}$ as a teacher network, focusing on reducing the distillation loss $d$ according to a distillation loss.

            Recalculate Distillation Loss $d$.

        Unfreeze all the reset neurons in Neural Network $f_\theta$.

        **if** $p + \gamma < 1$ **then**

            $p = p + \gamma$;

        **else**

            $p = 0$;

---

### A.3 ALGORITHM

In this section, we present two novel algorithms designed to achieve sustainable plasticity in neural networks: Sustainable Backup Propagation (SBP) and Plastic Proximal Policy Optimization (P3O). These algorithms address the challenge of maintaining neural network plasticity over extended periods.

#### A.3.1 SUSTAINABLE BACKUP PROPAGATION (SBP)

SBP is a general framework designed to maintain the plasticity of neural networks over extended periods. It incorporates a neuron regeneration mechanism into standard backpropagation, creating a sustainable learning process. The key components of SBP are:

- **Cycle Reset:** Periodically reinitializes a portion of neurons to prevent overspecialization.

- **Inner Distillation:** Utilizes a temporary model as a teacher network to recover essential knowledge after neuron reset.

This approach allows neural networks to maintain plasticity indefinitely, continually adapting to new information without catastrophic forgetting.

The SBP algorithm implements the cycle reset and inner distillation mechanisms. The reset rate $\gamma$ determines the proportion of neurons reset in each cycle, while the distillation process ensures that essential knowledge is retained after each reset.

### A.3.2 PLASTIC PROXIMAL POLICY OPTIMIZATION (P3O)

P3O is a concrete implementation of the SBP framework within the context of reinforcement learning, specifically tailored for the Proximal Policy Optimization (PPO) algorithm. A specialized distillation function, DKL, is designed for PPO, ensuring effective knowledge transfer in policy space. P3O demonstrates how the general SBP framework can be applied to specific machine learning paradigms, in this case, enabling sustainable plasticity in reinforcement learning policies.

Both algorithms represent a significant step towards creating AI systems that can learn continuously and adapt to changing environments without losing previously acquired knowledge. They offer a promising approach to overcoming the limitations of traditional neural network training methods, particularly in scenarios requiring long-term learning and adaptation.

---

**Algorithm 2** Plastic PPO(P3O)

---

Policy $\pi_\theta$, Temporary Policy $\pi_{\text{tmp}}$, Reset Rate $\gamma$, Training Steps $T$, Reset Frequency $F$, Reset Index $p = 0$, Distillation Threshold $\tau$, Distillation Loss $d =$ None. **for** $t = 1$ *to* $T$ **do**

    Update Policy $\pi_\theta$ with regular policy gradient.

    **if** $t \mod F = 0$ **then**

        Copy the weights of Policy $\pi_\theta$ to Temporary Policy $\pi_{\text{tmp}}$.

        **for** *each layer $L$ of the Network* **do**

            Let $l =$ neurons of layer $L$. Reinitialize input weights of neuron $i$ in layer $L$: $i \in [p \cdot l : (p + \gamma) \cdot l]$.

        Freeze all the reset neurons in Policy $\pi_\theta$.

        **while** $d > \tau$ *or* $d =$ *None* **do**

            Update Policy $\pi_\theta$ using Temporary Policy $\pi_{\text{tmp}}$ as a teacher network based on Equation 4.

            Update Distillation Loss $d$.

        Unfreeze all the reset neurons in Policy $\pi_\theta$.

        **if** $p + \gamma < 1$ **then**

            $p = p + \gamma$

        **else**

            $p = 0$

---

P3O adapts the SBP framework to the context of reinforcement learning. The key difference lies in the update mechanism policy gradient and the specialized distillation function (Equation 4) designed for policy space.

Both SBP and P3O represent significant advancements in creating AI systems capable of continuous learning and adaptation. By incorporating neuron regeneration and knowledge distillation, these algorithms offer a promising approach to overcoming the limitations of traditional neural network training methods, particularly in scenarios requiring long-term learning and adaptation to changing environments.

### A.4 ABLATION

#### A.4.1 ABLATION STUDY OF ACTIVATION

To comprehensively evaluate our algorithm's capability in maintaining plasticity, we conducted additional experiments across different activation functions, recognizing that neural networks exhibit

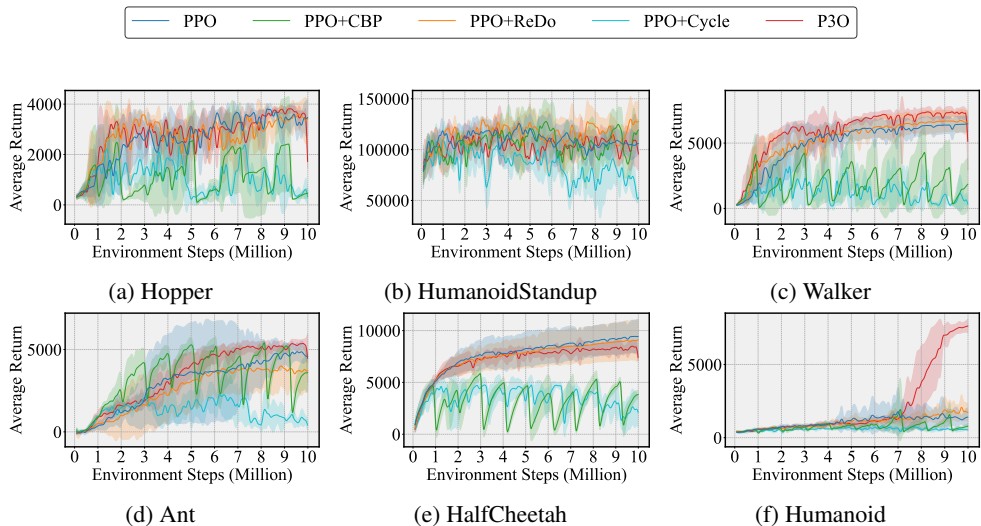

Figure 11: Performance Comparison of Different Reset Strategies with ReLU Activation in MuJoCo Environments.

varying characteristics under different activation schemes. We maintained the same parameter settings as shown in Table3 across all six environments.

1. **Cross-Activation Function Analysis:**
   - Our method demonstrates consistent performance improvements with both ReLU and tanh activation functions
   - The magnitude of improvement remains comparable between ReLU and tanh conditions

2. **Comparative Robustness:**
   - Baseline methods show strong performance with ReLU activation
   - However, these methods exhibit significant performance degradation with tanh activation
   - Our approach maintains stable performance across both activation functions

The results demonstrate that our algorithm exhibits superior robustness compared to existing methods, as it maintains consistent effectiveness regardless of the choice of activation function, showing that its plasticity-maintaining capabilities are not limited to specific activation functions but rather represent a more general and robust solution.

**Plasticity Exploration**

To better investigate the PPO algorithm, we chose the tanh activation function based on previous research experience. However, when attempting to use dormancy rate as a measure of plasticity, we made an unexpected discovery. In networks using tanh activation, we found a correlation between the magnitude of the activation function outputs and the size of the weights. Inspired by the neuron activation analysis in ReDo, we directly use the absolute value of neuron activations as the neuron scoring metric, since the activation values (tanh) are bounded in [-1,1]. This score reflects the activity level of each neuron, and we calculate the average score across all neurons to measure the overall network activation. As shown in Fig. 15 and 5, our analysis reveals a clear correlation between the overall magnitude of activation values and the weights.

To address this concern, we conducted additional experiments with ReLU activation functions, with results shown in Fig. 11, and calculated the dormant ratio using a threshold of 0.1 12. The dormancy rate curves closely align with performance variations - lower dormancy rates correlate with higher performance. Our method consistently maintains lower dormancy rates across most environments, following trends similar to those observed in weight and gradient norms (Fig. 16 and Fig. 13). However, the activation norm shows a more nuanced relationship. Comparing Fig. 15 and Fig.

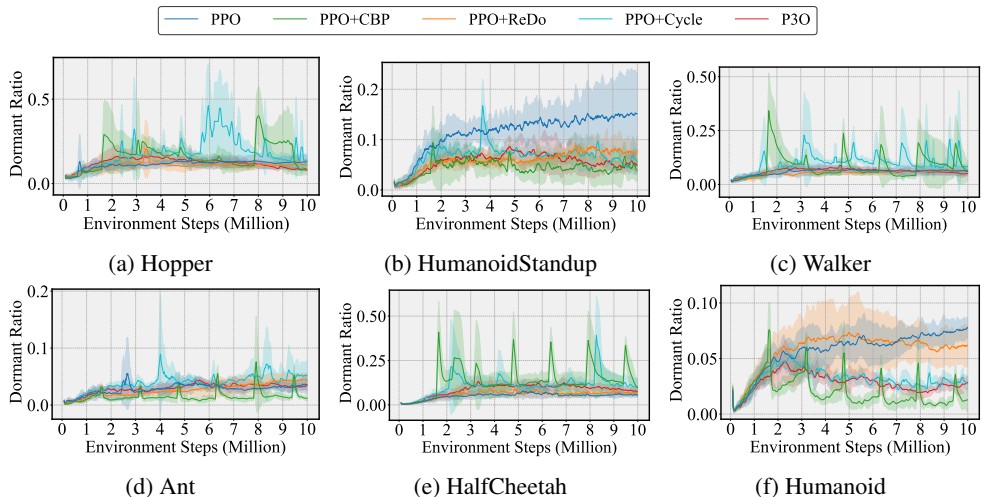

Figure 12: Dormant Ratio (Threshold=0.1) with ReLU Activation across MuJoCo Environments.

14 reveals that both extremely high and low activation norms can be problematic. Our method consistently maintains stable activation norms across environments - approximately 0.7 for tanh and 0.3 for ReLU activations. This suggests that maintaining activation values within specific, activation function-dependent ranges might be crucial for plasticity, though this hypothesis requires further investigation. Therefore, weight and gradient norms can effectively indicate neural network plasticity and demonstrate the effectiveness of our algorithm.

To further evaluate plasticity across methods, we visualized gradient covariance matrices across different environments. In Walker and HalfCheetah environments (Fig. 29, 28) where baseline and Redo achieve similar performance to P3O, the gradient correlations in P3O are notably lower than the other two methods, indicating better plasticity. In the Humanoid environment (Fig. 30), while other methods show largely uncorrelated neuronal patterns, P3O demonstrates richer neuron interactions, suggesting a more effective utilization of the neural network capacity - an advantage we attribute to knowledge distillation.

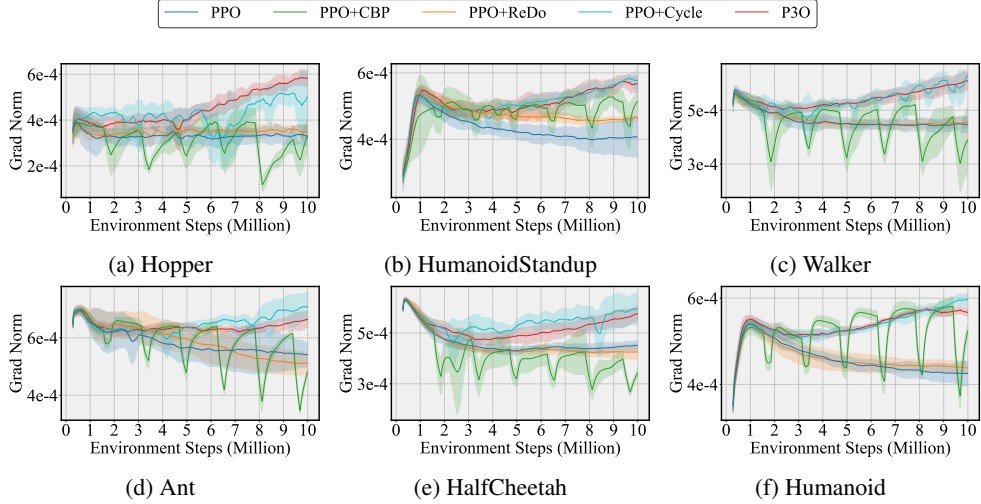

Figure 13: Grad Norm of Actor Network with ReLU Activation. Higher gradient norm indicates greater plasticity.

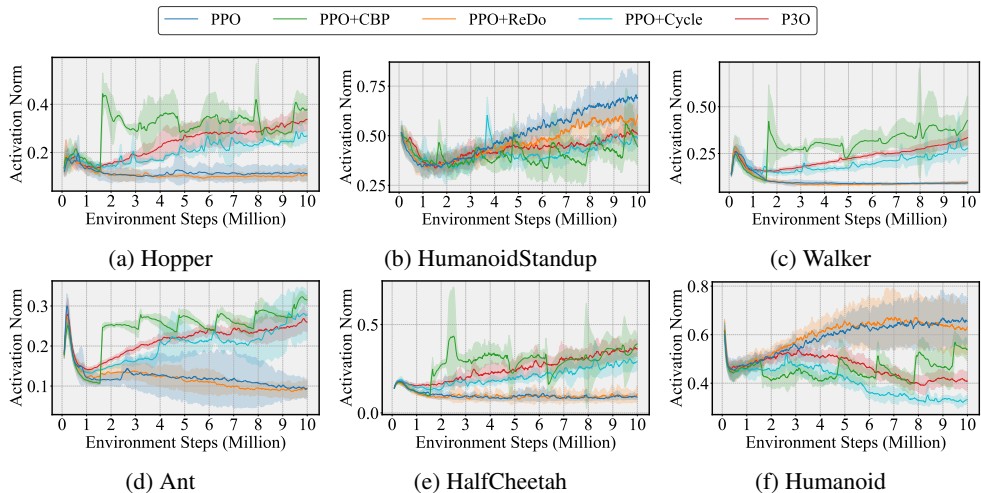

Figure 14: Activation Norm with ReLU Activation across MuJoCo Environments.

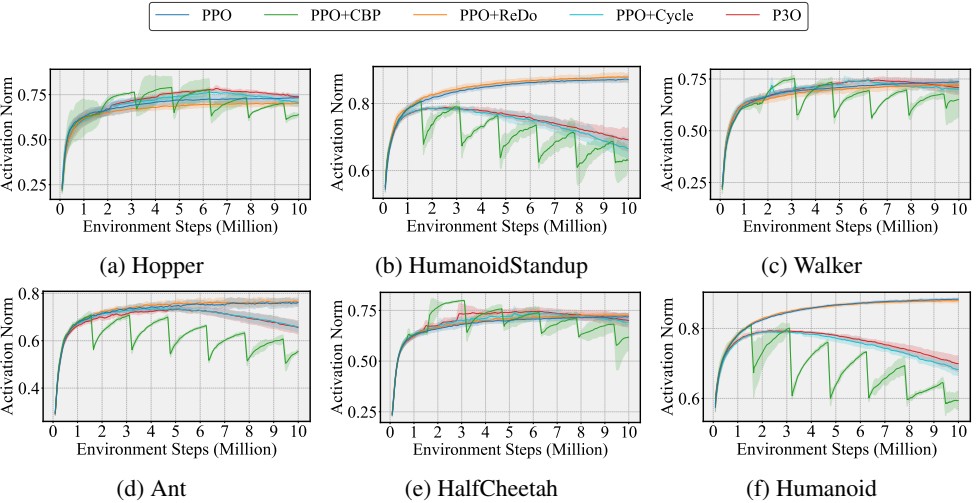

Figure 15: Activation Norm of tahn across six MuJoCo environments.

### A.4.2 ABLATION STUDY OF **RESET RATE**

To investigate the impact of reset rates across different environments, we conducted a series of ablation experiments. The results, illustrated in Figure 17, yield several significant observations:

- **Lower Reset Rates:** In most environments, lower reset rates generally yielded better performance. This suggests that retaining a larger portion of previously learned information is beneficial for many tasks.

- **Higher Reset Rates:** In some specific environments, such as the HumanoidStand task, higher reset rates (up to 0.5) produced superior results. This indicates that for more complex tasks, a significant portion of previously learned information may become obsolete, necessitating a higher reset rate to discard outdated knowledge.

- **Full Network Reset:** Interestingly, in environments like Walker and Half Cheetah, resetting the entire network outperformed resetting only half of it. This phenomenon could be attributed to the quality of newly sampled data in these environments. Learning from scratch with fresh data may lead to higher efficiency, especially in environments where performance is already satisfactory. This observation could potentially serve as evidence for the impact of primacy bias on learning.

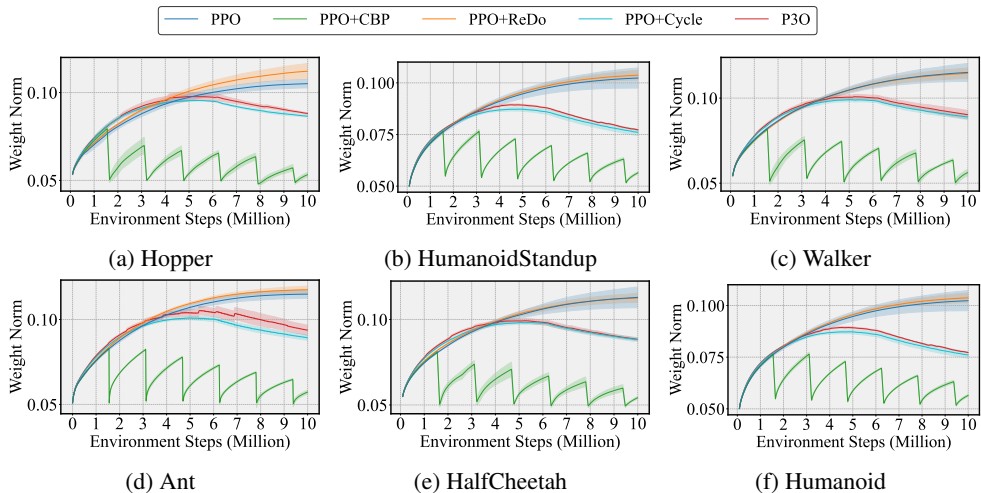

Figure 16: Weight Norm of Actor Network with ReLU Activation. Lower weight norm indicates greater plasticity.

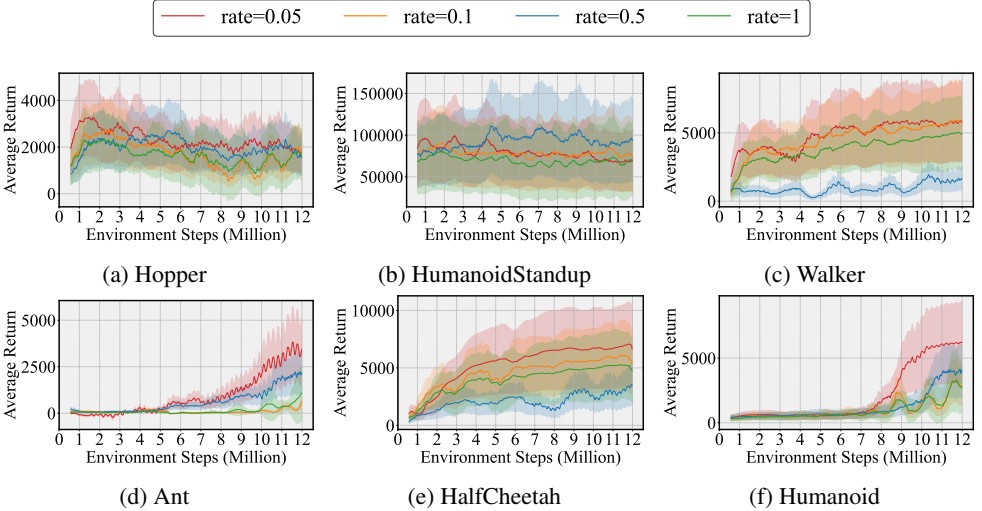

Figure 17: Ablation of Per-Reset Network Reset Ratio

- **Stability vs. Performance:** While higher reset rates showed improved performance in certain scenarios, they also introduced greater volatility compared to our default parameter of 0.01. This trade-off between performance and stability needs to be carefully considered.

These findings underscore the importance of task-specific tuning for reset rates. The optimal reset rate can vary significantly depending on the complexity of the task, the quality of sampled data, and the desired balance between performance and stability. Future work could explore adaptive reset rate mechanisms that automatically adjust based on task characteristics and learning progress.

### A.4.3 ABLATION OF RESET FREQUENCY

Fig. 18 presents our comprehensive analysis of various reset frequencies, specifically examining intervals of 20000, 40000, 80000, and 100000 steps. Our results did not reveal a clear, universally applicable pattern across different environments. Instead, we observed that the performance of neuron regeneration at different frequencies varied significantly depending on the specific task environment. This variability suggests that the optimal reset frequency is highly task-dependent and requires case-by-case analysis. The lack of a consistent trend across environments highlights the complexity

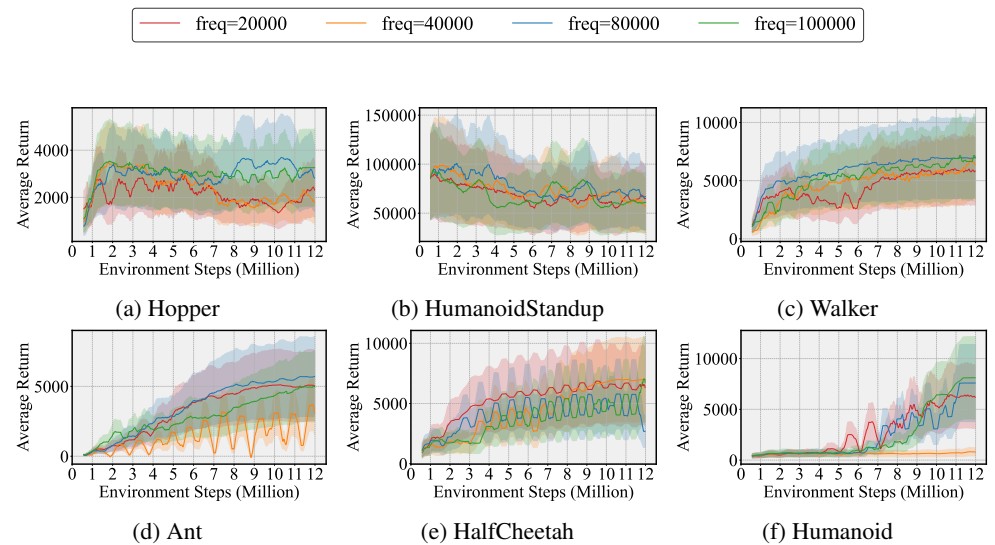

Figure 18: Ablation of Reset Frequency

of determining an ideal neuron regeneration cycle. Our findings indicate that the reset frequency is a nuanced parameter that cannot be universally prescribed. This complexity underscores the need for further research to better understand the relationship between task characteristics and optimal reset frequencies. Future studies should aim to develop a more comprehensive framework for determining appropriate reset frequencies based on specific task attributes and learning dynamics. The challenge of identifying optimal reset frequencies represents an important area for ongoing investigation in the field of sustainable neural plasticity.

### A.4.4 ABLATION OF **RESET RATIO**

In this section, we investigate the impact of different reset ratio through ablation studies to validate the necessity of cycle reset.

We conducted experiments by restricting the reset range to the initial x% (x  0.2, 0.5, 0.8) of neurons in the network throughout training. The resulting weight and gradient variations are shown in Fig. 20 and Fig. 19, respectively.

Our experiments reveal a clear correlation between reset ratio and network characteristics:

1. Higher reset ratio lead to smaller weight magnitudes across the network

2. Networks with higher reset ratio maintain larger gradient norms throughout training

These findings support our hypothesis about the relationship between reset mechanisms and network plasticity - higher reset ratio are crucial for maintaining the network in a more plastic state, as evidenced by both the weight magnitudes and gradient characteristics. Building upon this insight, our cycle reset mechanism systematically resets neurons in order and ensures that the longest-surviving neurons are reset in each cycle, providing a principled approach to address the issues of weight magnitude growth and gradient diminishing. The experimental results validate that this mechanism is both well-founded and essential for maintaining network plasticity throughout training.

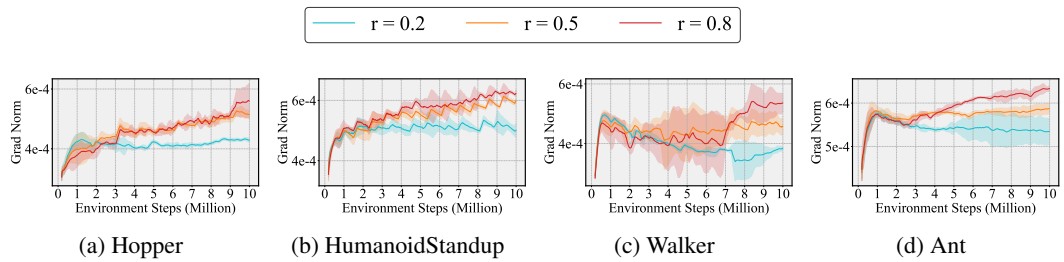

Figure 19: Grad Norm: Total Network Reset Ratio over Complete Training

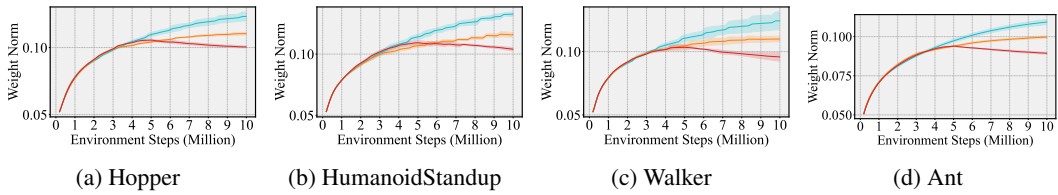

Figure 20: Weight Norm: Total Network Reset Ratio over Complete Training

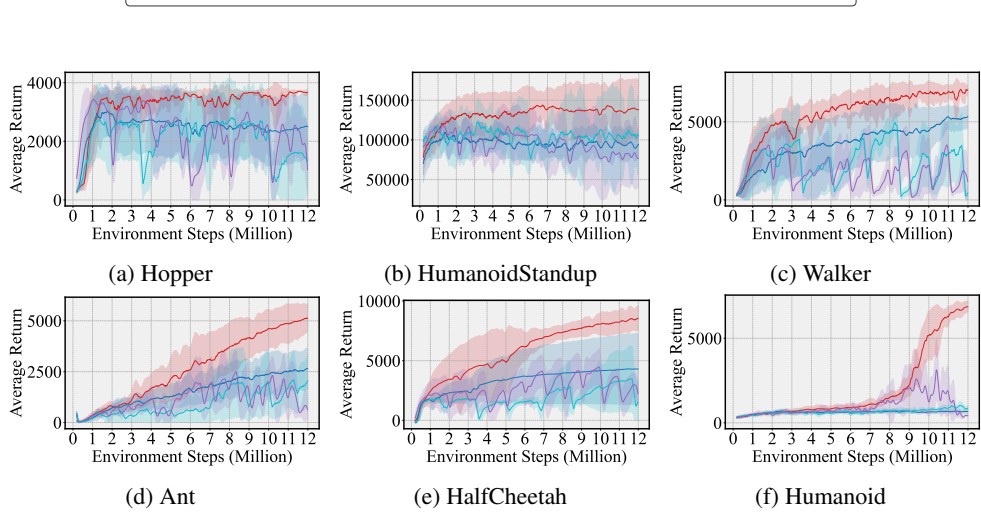

Figure 21: Performance Comparison with Recovery Training (RT)

### A.4.5 INNER DISTILLATION VERSUS EXTRA TRAINING

To rigorously evaluate the necessity of our distillation mechanism, we conducted comparative experiments between inner distillation and simple extra training after cycle reset. We implemented a recovery-based training protocol where the network was allowed additional training epochs post-reset until reaching a reward threshold of 100, before proceeding to the next training cycle. This

Table 5: Comparison of Distillation Epochs and Extra Recovery Epochs

|  | Hopper | Humanoid Stand | Walker | Ant | HalfCheetah | Humanoid |
|---|---|---|---|---|---|---|
| Distillation Epochs | 597.66 | 638.19 | 1536.80 | 7698.40 | 2548.40 | 6525.25 |
| Recovery Epochs | 809.75 | 2731.60 | 524.00 | 700.20 | 253.20 | 485.25 |

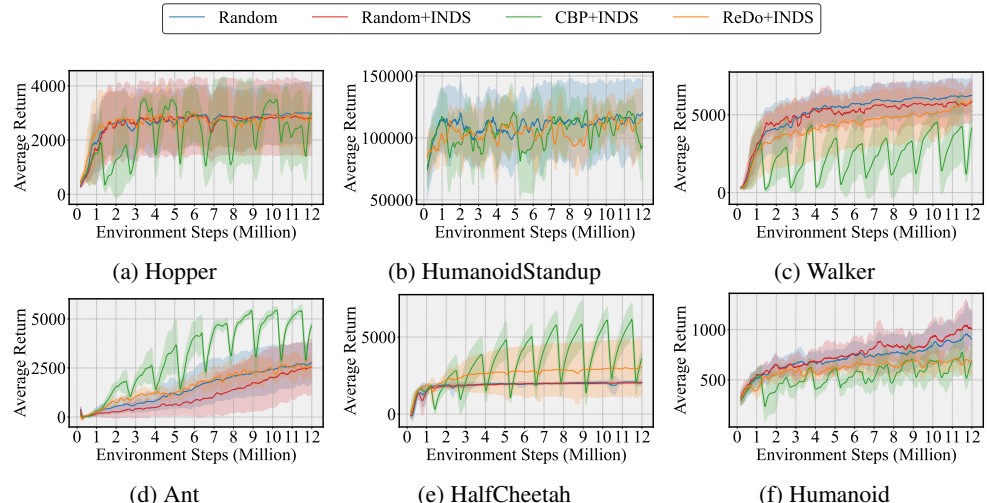

Figure 22: Baseline with Inner Distillation(INDS).

experimental design allows us to directly compare the effectiveness of distillation versus simple training recovery in mitigating knowledge loss caused by reset. Our experimental results, as shown in Fig. 21, reveal that recovery to moderate reward levels through extra training requires less computational cost and performs slightly better than cycle reset alone. However, while extra training is efficient for basic recovery, it fails to generate substantial performance improvements, indicating that the ability to recover previous performance does not guarantee continued learning progress. In contrast, our distillation mechanism demonstrates superior capabilities in sample efficiency, network plasticity utilization, and sustained performance improvements beyond simple recovery. These findings emphasize that while simple retraining cannot fully recover lost knowledge, our inner distillation mechanism effectively preserves and restores knowledge while maximizing learning efficiency and leveraging the network's plastic potential.

**Baseline with Inner Distillation**

To further explore the value of knowledge preservation through distillation, we conducted experiments combining various baselines with inner distillation. As illustrated in Figure 22, we observed that the performance improvements brought by distillation were limited for random resets and ReDo. However, for CBP, distillation sometimes yielded beneficial improvements, notably in the Ant environment where positive gains were observed. Despite these improvements, the overall learning process remained unstable. It's worth noting that we used default parameters in these experiments, which may have limited the full integration of our algorithm with CBP.

These findings lead to an encouraging conclusion: distillation can be effectively combined with other reset strategies, serving as a mechanism for knowledge retention in new neuronal regeneration paradigms. While the results are promising, they also highlight the need for further research to optimize the various parameter settings. This refinement process is crucial for fully realizing the potential of combining distillation with neuronal reset strategies and developing more robust and effective learning algorithms.

## A.5 SAC WITH SBP

To further validate the generality of our framework, we integrated SBP into SAC and evaluated it across four Mujoco environments, as shown in Fig. 23. We compared both redo and cycle reset approaches with the hyperparameter configurations detailed in Table 6. Our results demonstrate that SBP consistently improves SAC's performance. The analysis of dormant ratio (Fig. 24), gradient norm (Fig. 27), weight norm (Fig. 26), and activation norm (Fig. 25) reveals trends similar to those observed in PPO: lower dormancy rates, larger gradients, smaller weights, and activation values maintained within stable ranges. However, we believe these improvements represent only a fraction of SBP's potential benefit to SAC, particularly considering that our current implementation, which

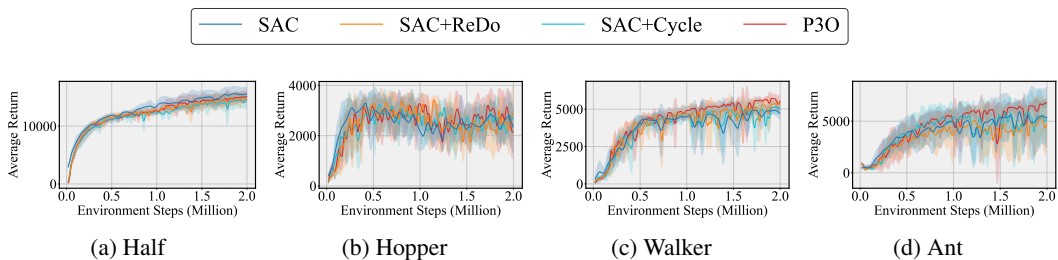

(a) Half  (b) Hopper  (c) Walker  (d) Ant

Figure 23: Performance Comparison of SAC across four MuJoCo environments.

randomly samples just 1% (8,192 samples) from the replay buffer for inner distillation, already achieves significant performance gains. The current approach, while effective, leaves substantial room for exploring more sophisticated sampling strategies to better utilize the rich information available in off-policy settings.

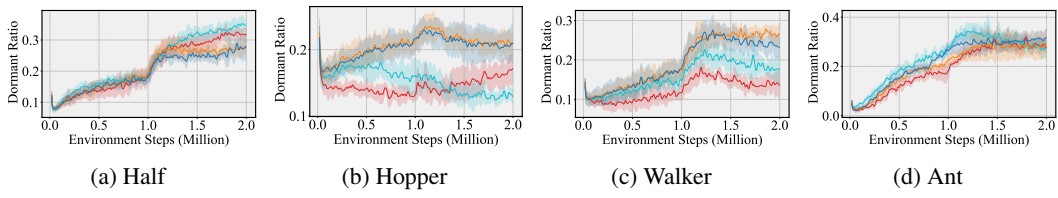

(a) Half  (b) Hopper  (c) Walker  (d) Ant

Figure 24: Dormant Ratio (Threshold=0.1) of SAC across four MuJoCo environments

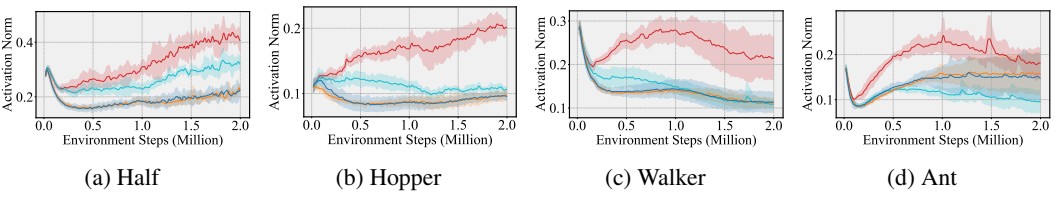

(a) Half  (b) Hopper  (c) Walker  (d) Ant

Figure 25: Activation Norm of SAC across four MuJoCo environments.

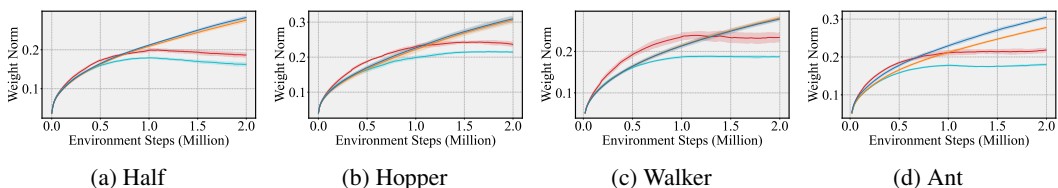

(a) Half  (b) Hopper  (c) Walker  (d) Ant

Figure 26: Weight Norm of SAC across four MuJoCo environments.

Table 6: Hyperparameter Configuration of SAC with SBP

| Parameter | Value |
| --- | --- |
| Reset Frequency | 10,000 steps |
| Reset Percentage | 0.01 |
| Alpha Value | 0.8 |
| Replay Buffer Size | 1 million samples |
| Distillation Buffer Size | 8,192 samples |

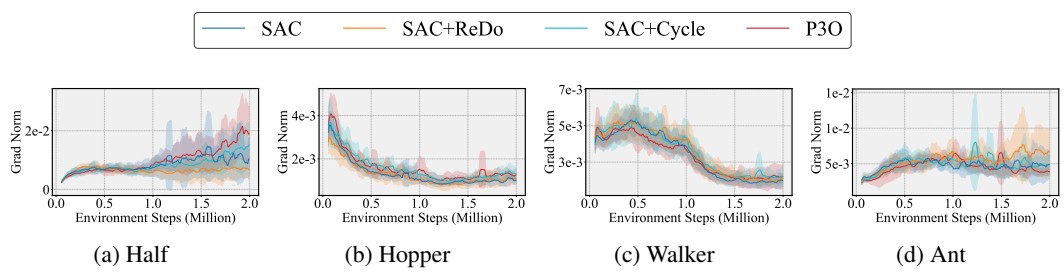

Figure 27: Grad Norm of sac across four MuJoCo environments.

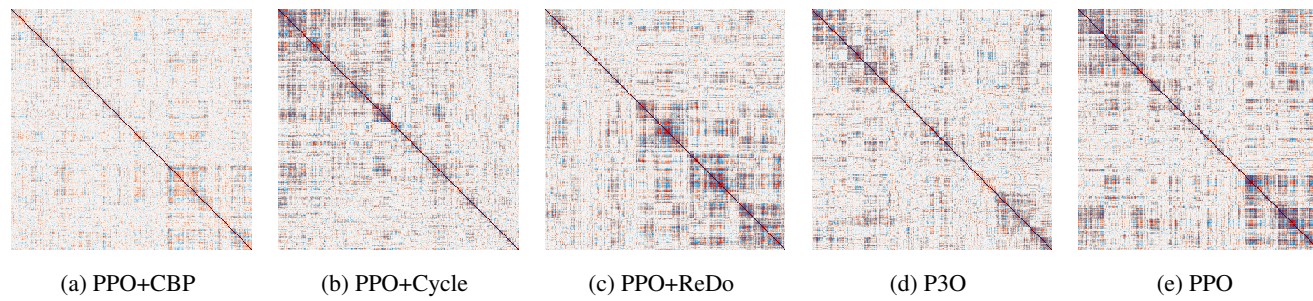

Figure 28: Gradient Correlation of Walker-2d across Different Algorithms.

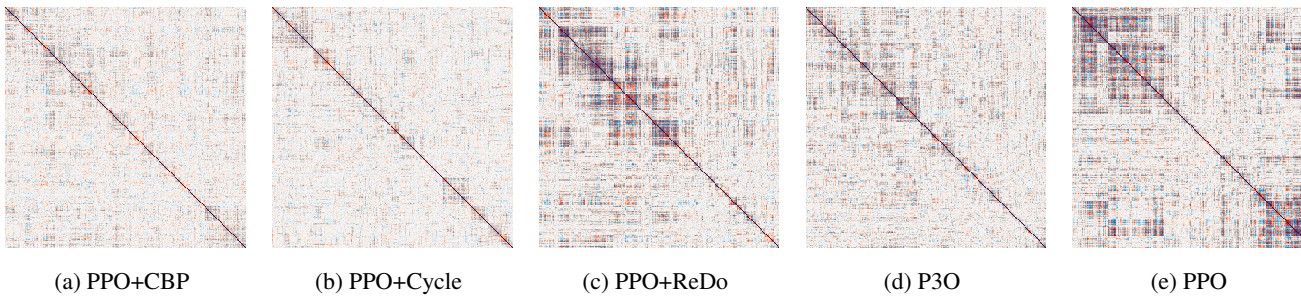

Figure 29: Gradient Correlation of HalfCheetah across Different Algorithms

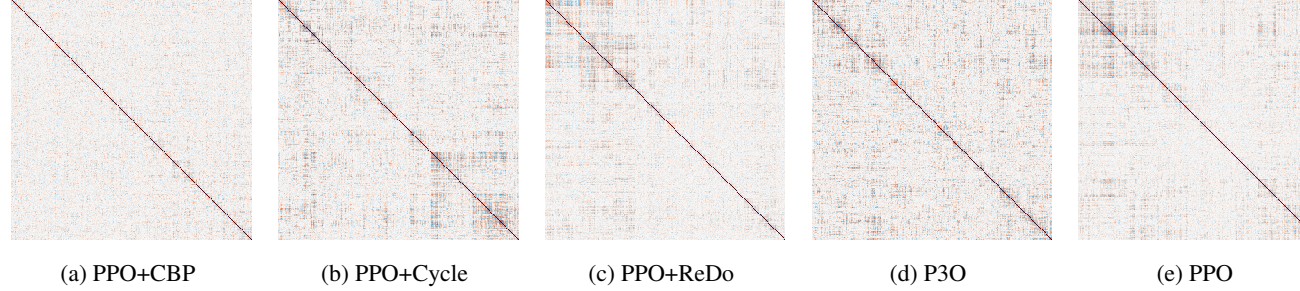

Figure 30: Gradient Correlation of Humanoid across Different Algorithms. Dark red represents strong negative correlation, while dark blue indicates strong positive correlation.

