# OpenReview forum: "Stay Hungry, Keep Learning: Sustainable Plasticity for Deep Reinforcement Learning"
_ICLR.cc/2025/Conference — Submitted to ICLR 2025_

### Official Review · Reviewer_CXrK · 2024-10-30

**Soundness:** 2
**Presentation:** 3
**Contribution:** 2
**Rating:** 5
**Confidence:** 5

**Summary:**

This paper proposes a straightforward and reasonable method, to improve the performance of reset, which is one of the most reliable methods to mitigate plasticity loss, in Gym-mujoco and DMC.

**Strengths:**

This paper is well-written and has clear insight.

The method is introduced in a reasonable and theatrical way.

The experimental scenarios are diverse and convincing.

**Weaknesses:**

1. Regarding the experiments with DMC, I couldn't find how you built the module for encoding image input. Could you describe it in detail?

2. Using the original PPO is rarly in DMC testing. Because it is difficult to learn effective policies, it cannot provide algorithm effectiveness support. Examples are Dog Walk and Hopper Hop in the paper. The learning curve shows that none of the methods learned an effective strategy (the highest score is below 0.1). This only adds to my doubts about P3O's performance.

3. The full paper does not say how many seeds were used for each experiment. In addition, the large difference in the variance shading of each curve in Figure 5 calls into question the fairness of its experimental design (0 variance for PPO+CBP).

4. This paper lacks the comparison of the methods on plasticity evaluation metrics, such as covariance metric [1].

5.  Most of the results on Mujoco in Fig 14 show that the proposed method performs significantly worse than other methods on mainstream plasticity evaluation metrics. The reasons given are confusing and reinforce my suspicions about the mismatch between P3O and motivation. I would advise authors to replace the middle layer activation function with ReLU and re-record the metric and learning curve (generally it doesn't influence much). It's very limiting if your approach works on tanh activation functions.

[1]  Ceron, Johan Samir Obando, Aaron Courville, and Pablo Samuel Castro. "In value-based deep reinforcement learning, a pruned network is a good network." Forty-first International Conference on Machine Learning.

**Questions:**

Please see the above suggestions.

---

> ### Author Response · Authors · 2024-11-26
>
> We sincerely thank the reviewer for their insightful comments and valuable suggestions. Below we address the concerns raised:
>
> Response to 1 and 2
>
> We used the state-vector version of DMC rather than image-based inputs, as CleanRL's PPO benchmark results[^1] show that PPO can achieve moderate performance with vector states. We specifically chose environments where PPO demonstrates learning capability while leaving room for improvement. This experimental design ensures the tasks are learnable (as evidenced by baseline performance) while providing sufficient headroom to demonstrate the benefits of enhanced plasticity. We will clarify this experimental setup in the revision. We apologize for any confusion caused by the heavily smoothed visualization of the baseline performance. We have updated Fig. 9 to show the raw baseline data, which more accurately reflects the actual baseline results with returns consistently reaching 4-5 and occasional peaks above 10. This revision provides a clearer representation of the baseline algorithm's true performance.
>
> Response to 3
>
> We acknowledge that Fig.  5 inadvertently combined results from different parameter settings. We have regenerated the figure using consistent settings across all methods, with data from 5 random seeds for each experiment. The revised figure will be included in the updated manuscript along with explicit documentation of the seed count.
>
> Response to 4
>
> Following the reviewer's suggestions, we have expanded our evaluation in two key aspects. First, we incorporated two additional plasticity metrics: neuronal dormant ratio and gradient covariance. Second, we conducted experiments replacing tanh with ReLU in middle layers. The new results, detailed in the Appendix A.4.1, demonstrate our method's effectiveness across different activation functions while providing further validation of its plasticity properties.
>
> Response to 5
>
> We apologize for the presentation issues in our manuscript. Similar to the previously acknowledged error in Fig. 5, Fig. 14 was incorrectly labeled and contained results from mismatched experimental configurations. Additionally, we failed to clarify that the figure shows activation norm data, where we used absolute values of tanh outputs as neuron scores. We have now replotted all metrics with consistent experimental settings and conducted additional validation with ReLU activation functions in the Appendix A.4.1, and changed Fig. 14 to Fig. 15. The new results demonstrate our method's consistency with standard plasticity metrics when using ReLU.
>
> [^1]: https://wandb.ai/openrlbenchmark/openrlbenchmark/reportlist

---

> > ### Author Response · Authors · 2024-12-03
> >
> > Dear Reviewer CXrK,
> >
> > I hope this message finds you well. This is a gentle reminder regarding our manuscript review. We deeply appreciate your invaluable comments and thorough feedback that helped improve our paper. As the rebuttal phase is ending soon, we would be grateful if you could review our responses and consider adjusting the paper's rating. We believe we have carefully addressed all your concerns in our revision.
> >
> > Thank you very much for your time and support throughout this review process.

---

### Official Review · Reviewer_Ru1A · 2024-10-31

**Soundness:** 2
**Presentation:** 3
**Contribution:** 2
**Rating:** 5
**Confidence:** 4

**Summary:**

This paper introduces the concept of "neuron regeneration" - a mechanism designed to maintain neural network plasticity while preserving stability in deep reinforcement learning systems. The core contribution is the Sustainable Back Propagation (SBP) framework, which implements neuron regeneration through two complementary mechanisms: cycle reset and inner distillation. The cycle reset component periodically reinitializes selected neurons, while inner distillation ensures the preservation of essential knowledge during the reset process.
The authors integrate SBP with Proximal Policy Optimization (PPO) to create Plastic PPO (P3O), demonstrating the practical application of their framework. The experimental results show that P3O achieves superior performance in both standard single-task environments and non-stationary scenarios, suggesting effective maintenance of plasticity without compromising stability. The empirical evaluation spans multiple environments, including MuJoCo benchmarks and custom non-stationary tasks, providing evidence for the framework's effectiveness in addressing the plasticity-stability trade-off in deep reinforcement learning.

**Strengths:**

1. The key innovation lies in the inner distillation mechanism, which effectively mitigates the potential negative impacts of reset. This approach represents a promising solution to preserve critical knowledge during the reset process.

2. The paper presents a novel neuron-level reset approach that periodically refreshes different neurons on a cyclical basis, rather than specifically identifying neurons for reset (as in ReDo). While this represents an interesting direction that warrants further investigation (see weaknesses and questions), it introduces a fresh perspective on maintaining network plasticity.

3. The experimental validation is comprehensive, encompassing both standard control tasks and non-stationary scenarios. This breadth of evaluation, particularly the inclusion of environments with changing dynamics, strengthens the empirical support for the proposed method.

4. The use of weight norm and gradient norm as analytical metrics provides valuable insights into network characteristics. These measurements offer a quantitative lens for understanding how the proposed method affects network behavior and learning dynamics.

**Weaknesses:**

1. The necessity of introducing "neuron regeneration" as a new concept is questionable, as it heavily overlaps with the established plasticity-stability trade-off. The proposed approach essentially describes enhanced plasticity without stability degradation, which could be framed within existing theoretical frameworks.

2. Several terminological choices appear disconnected from their underlying methods. The title "Stay Hungry, Keep Learning" shows little relation to cycle reset and inner distillation. The framework name "Sustainable Back Propagation (SBP)" is potentially misleading as it doesn't directly involve modifications to backpropagation. Moreover, the inconsistent reference to SBP as "Sustainable Brain Plasticity" in the abstract raises concerns about terminology precision.

3. While neuron-level cyclic reset is a key component, the paper lacks sufficient justification for its superiority over targeted reset approaches. Although ReDo comparison is included, a more comprehensive analysis is needed, particularly with aligned reset frequency and magnitude between methods. This is crucial given the central role of the cyclic reset strategy in the paper's contribution.

4. Despite the paper's ambitious claim of achieving "Sustainable Plasticity", the choice of metrics (weight norm and gradient norm) as indirect indicators is less convincing than currently preferred measures in the community, such as fraction of active units or dormant ratio. These more direct plasticity metrics would provide stronger support for the paper's claims.

5. A significant limitation is the framework's validation being restricted to PPO. To demonstrate the effectiveness and generality of SBP, testing on DQN-based methods and other off-policy actor-critic algorithms is essential. This broader validation would substantially strengthen the paper's contribution to the field.

**Questions:**

I believe each weakness identified above requires careful consideration and substantive response from the authors. A clear path to acceptance would involve addressing these concerns with concrete solutions and additional empirical evidence where necessary. I am open to further discussion during the rebuttal period and would be willing to revise my assessment positively if the authors can effectively address these issues.

For acceptance, I would particularly emphasize the need for:
- Better justification of the neuron regeneration concept
- More comprehensive comparison with targeted reset approaches
- Additional plasticity metrics
- Broader algorithmic validation

I look forward to the authors' response and am prepared to engage in constructive dialogue during the rebuttal period. A thorough addressing of these concerns would warrant a more favorable evaluation.

---

> ### Author Response · Authors · 2024-11-26
>
> We sincerely thank the reviewer for their insightful comments and valuable suggestions. Below we address the concerns raised:
>
> Response to 1
>
> While the plasticity-stability trade-off provides broad concept, our proposed concept of "neuron regeneration" serves two essential purposes that justify its introduction:
>
> 1. Precision and Specificity: Unlike the broader plasticity-stability trade-off concept, "neuron regeneration" focuses precisely on neuron-level phenomena and mechanisms. By explicitly describing operations at the individual neuron level, this term provides a more concrete and specific way to discuss neural network plasticity. This neuron-level specificity helps researchers better understand and analyze the actual mechanisms involved in maintaining network plasticity.
>
> 2. Conceptual Clarity and Biological Inspiration: The term "neuron regeneration" deliberately draws parallels with biological processes, making it more intuitive for researchers to understand the purpose and mechanisms of our approach. This biological analogy helps convey how neural networks can maintain stability while enabling continuous learning through targeted neuron-level interventions.
>
> We refined the definition to highlight neuron-level operations. This more precise terminology contributes to the field by focusing specifically on neuronal mechanisms involved in maintaining network plasticity and stability, rather than relying on broad concepts.
>
> Response to 2
>
> The title "Stay Hungry, Keep Learning" intentionally captures our method's core philosophy - maintaining continuous learning capability and avoiding saturation, which directly aligns with our cycle reset and inner distillation mechanisms. Just as biological systems maintain learning capacity through constant renewal, our method sustains the network's ability to acquire new knowledge.
>
> Regarding the framework name, we agree that "Sustainable Back Propagation" might not fully reflect our method's essence. We have revised it to "Sustainable Backup Propagation (SBP)", which more accurately describes our approach of maintaining sustainable learning through knowledge backup and support mechanisms. This new terminology better aligns with our method's core components: the cycle reset that provides sustainable learning capability and the knowledge distillation that serves as a backup and support system for preserving critical information.
>
> We also acknowledge that there was a typographical error in the abstract where "Sustainable Back Propagation (SBP)" was incorrectly written as "Sustainable Brain Plasticity". This was an oversight during manuscript preparation. Throughout the revised paper and in all our technical discussions, we consistently use and define SBP as "Sustainable Backup Propagation", which accurately reflects the nature of our proposed method. We have corrected these terminology issues in the revised manuscript to maintain precision and consistency.
>
> Response to 3
>
> We conducted experiments comparing our cyclic reset strategy with targeted approaches. Through analysis of weight norms (Fig. 5) and gradient norms (Fig. 6), we demonstrate that cyclic reset is the primary factor in reducing weight magnitudes and maintaining gradient flow, while distillation serves as a knowledge preservation mechanism that maintains learning efficiency and ensures performance improvement. Additionally, we validate the necessity of resetting all neurons through comparative experiments that examine different reset ratios. The results consistently show that higher reset ratios correlate with both smaller weight magnitudes and larger gradient values, supporting our hypothesis about the relationship between reset coverage and network plasticity. This relationship demonstrates why full neural reset is more effective than partial reset strategies. Details have been added to the Appendix A.4.4.
>
> Response to 4
>
> Beyond the indirect metrics of weight and gradient norms, we have now incorporated direct plasticity measurements - the neuronal dormant ratio and activation norm. These additional metrics further validate our framework's effectiveness in maintaining network plasticity. Details are provided in the Appendix A.4.1.
>
> Response to 5
>
> Our focus on PPO was strategic: on-policy algorithms face unique plasticity challenges due to their small online buffer and sensitivity to data distribution changes. This makes them an ideal testbed for our framework - success in this challenging setting would suggest broader applicability. Following the reviewer's suggestion, we validated this hypothesis with SAC experiments, which demonstrated performance improvements even with limited distillation buffer sizes. These results across both paradigms confirm our framework's effectiveness, while highlighting its particular value for on-policy methods where plasticity challenges are more acute. Details are provided in the Appendix A.5.

---

> > ### Author Response · Authors · 2024-12-03
> >
> > Dear Reviewer Ru1A,
> >
> > I hope this message finds you well. This is a gentle reminder about our manuscript. We sincerely appreciate your constructive feedback that has helped enhance our paper. As the discussion period is coming to an end, we would greatly appreciate if you could review our responses and consider updating the paper's rating. We have made dedicated efforts to address each of your concerns.
> >
> > Thank you for your valuable contribution to improving our work.

---

### Official Review · Reviewer_EKtA · 2024-10-31

**Soundness:** 4
**Presentation:** 3
**Contribution:** 4
**Rating:** 8
**Confidence:** 5

**Summary:**

A neural network’s ability to learn diminishes over time due to primacy bias and dead neurons. In order to maintain network plasticity, the authors propose a method of neuron regeneration that combines reset mechanisms and knowledge recovery. The approach, Sustainable Back Propagation, resets random neurons and trains the remaining neurons with knowledge/policy distillation from the pre-reset network. This approach is implemented with PPO as P3O and is tested against multiple RL environments.

**Strengths:**

This paper provides good framing and a clear picture of the issues in neuron plasticity. It provides good theoretical backing with neuron regeneration.

Descriptions are clear and concise, visuals are also clear and aid in understanding.

The approach improves on baseline PPO performance in all tested environments, beats the other methods in most experiments, and greatly surpasses the other methods in several experiments.

**Weaknesses:**

While you do an ablation of alpha values (for forward vs backward KL), you don’t test alpha=0.5, which represents a common way to turn KL divergence into a symmetrical distance function. You should test alpha=0.5, because if it is optimal, you wouldn’t need to explain the alpha-DKL as much, as symmetrical KL distance is well-established.

While the improved plasticity is clear with the sustained upward trend of the approach’s reward curves, it is unclear how to quantify plasticity. This may be out-of-scope for this paper, but a measure of plasticity could greatly strengthen the claims made. These could include analyzing the slope of the reward curve, the magnitude of the gradient, or perhaps there is an information theoretic angle to plasticity. You analyzed the magnitude of weights, but it is unclear if lower magnitude weights captures meaningful plasticity.

This is a minor gripe, but the explanation of PPO clipping in 2.1. is incorrect. The clip function does not simply limit the magnitude of policy updates. Instead, the clipping essentially sets the gradient to 0 and is used to prevent the model from learning a policy update that would take it further outside rt(theta) +/- epsilon. Using a similar explanation would be more accurate to the mechanics of PPO.

Minor formatting error: the PPO reference paper Schulman et al. 2017 appears twice in the Reference section.

**Questions:**

How can plasticity be calculated? Are lower magnitude weights always more plastic than higher magnitude weights?

---

> ### Author Response · Authors · 2024-11-26
>
> We sincerely thank the reviewer for their insightful comments and valuable suggestions. Below we address the concerns raised:
>
> Response to alpha
>
> Initially, we experimented with $\alpha=0.5$ (symmetric KL divergence), a theoretically well-established choice. However, our empirical results showed suboptimal performance in our setting. Through extensive ablation studies, we explored different $\alpha$ values to better balance the forward and reverse KL divergence terms. We have included comprehensive results comparing $\alpha=0.5$ with other $\alpha$ values in Fig. 10.
>
> Response to plasticity and weight magnitude
>
> Neural plasticity measurement remains an open challenge that can be approached through multiple metrics. Weight-based metrics suggest higher plasticity correlates with smaller weight magnitudes [1]. We focus on gradient norms as they directly indicate learning capacity and help identify plasticity loss through gradient vanishing. Additional metrics include activation patterns, where lower dormant ratio [2] suggests greater plasticity, and more recent approaches using covariance matrices [3]. While no single metric fully captures plasticity, examining these complementary measures together provides a more comprehensive understanding of network adaptability.
>
> Response to typo
>
> We thank the reviewer for this technical correction. We will revise Section 2.1 to more accurately describe PPO's clipping mechanism. The mechanism sets gradients to zero when $r_t(\theta)$ exceeds $[1-\epsilon, 1+\epsilon]$, thereby preventing policy updates rather than merely limiting their magnitude. We will also correct the duplicate citation of Schulman et al. 2017 in the References.
>
> ### References
>
> [1] Dohare et al. "Loss of plasticity in deep continual learning." Nature (2024)
>
> [2] Sokar et al. "The dormant neuron phenomenon in deep reinforcement learning." ICML (2023)
>
> [3] Obando et al. "In value-based deep reinforcement learning, a pruned network is a good network." Architecture (2024)

---

> > ### Comment · Reviewer_EKtA · 2024-11-27
> >
> > Thank you for your response.
> >
> > I agree that using multiple measures together would provide a stronger view of plasticity. Since you use both the weight and gradient norm, I am now convinced that your measure of plasticity is adequate.
> >
> > I still consider this paper to be in the "accept" category, though more solidly. I will increase the soundness and confidence score, but I will keep the overall score at 8.

---

> > > ### Author Response · Authors · 2024-11-28
> > >
> > > We sincerely thank you for your timely response and careful reading of our manuscript. Your valuable suggestions have helped us improve our paper. We are grateful for your recognition of our work and will continue our dedicated efforts.

---

### Official Review · Reviewer_Tisv · 2024-11-03

**Soundness:** 1
**Presentation:** 2
**Contribution:** 2
**Rating:** 3
**Confidence:** 4

**Summary:**

This paper targets plasticity loss in reinforcement learning. Specifically, the widely-used periodic resetting technique was used, which is re-named as "Cycle Reset" in this paper. The authors then adopted a knowledge distillation method to distill knowledge from policy before resetting to alleviate performance degradation from resetting. The authors applied the proposed method to PPO and conducted experiments in the MuJoCo and DMC benchmarks. Overall, the technical contribution of this paper is minor and the experimental setting is not convincing, which will be detailed below.

**Strengths:**

1. The authors introduce knowledge distillation into the resetting process of RL to alleviate performance drop after each resetting step, which shows good results in their experiments.

**Weaknesses:**

1. The technical contribution is minor.

1.1. First of all, I see no essential difference between the introduced "Cycle Reset mechanism" and the existing widely-used resetting mechanism [1]. It is inappropriate for the authors to claim this as their own contribution.

1.2. Secondly, the authors used an established knowledge distillation (KD) method in their distillation module, leaving the contribution of their method to be introducing KD into the resetting procedure.

2. The major research challenge this paper targets is the performance degradation after resetting. However, it has been well shown in previous works [1][2] that the reset model can quickly recover the lost ability after a few training steps, which makes the "performance degradation" not a significant problem from my point of view. On the other hand, the introduced KD module in this paper also involves additional cost in the overall RL procedure, which is not compared in the paper.

3. The experiment setting which only uses PPO is not convincing.

3.1. First of all, in [2], it has been revealed that the critic suffers much more than the actor w.r.t. plasticity loss, indicating that value-based or hybrid methods like DQN and SAC are more suited for plasticity loss study than policy-gradient methods like PPO. In Figure 4, PPO+Cycle also brings no benefit compared with PPO, showing that the proposed Cycle Reset does not help plasticity loss in this setting.

3.2. PPO is known for the difficulty in training, requiring potentially careful hyperparameter tuning. The results provided by the authors seem sub-optimal which may be due to this reason. Additionally, the results reported are highly likely to be unfair for previous methods due to a different experiment setting. In comparison, in [1], the performance of SAC / SAC+ReDo for HalfCheetah (MuJoco) can achieve a return over 10000, which is much higher than the reported on in Figure 4. And in [2], their proposed adaptive replay ratio can achieve a return of ~800 in Quadruped Run (DMC) which is also much higher than the reported result in Figure 9. Therefore, I highly recommend the authors to implement their methods on RL backbones used by previous works such as DQN, SAC, or DrQ.

[1] Sokar, Ghada, et al. "The dormant neuron phenomenon in deep reinforcement learning." International Conference on Machine Learning. PMLR, 2023.

[2] Ma, Guozheng, et al. "Revisiting Plasticity in Visual Reinforcement Learning: Data, Modules and Training Stages." The Twelfth International Conference on Learning Representations. 2024.

**Questions:**

1. Could the authors provide comparison on the extra cost introduced by (1) continually training the model after reset to recover the performance and (2) using knowledge distillation?

2. Could the authors provide experiment results on other RL methods other than PPO, such as DQN or SAC for a fair comparison with previous works on plasticity loss in RL?

---

> ### Author Response · Authors · 2024-11-26
>
> We sincerely thank the reviewer for their insightful comments and valuable suggestions. Below we address the concerns raised:
>
> Response to Weaknesses
>
> Response to  1.1
>
> Our Inner Distillation module represents a novel framework that extends significantly beyond traditional knowledge distillation.
>
> 1. First, while conventional KD transfers knowledge between separate teacher and student networks, our approach innovates by enabling fine-grained knowledge transfer between different components within the same network at the neuronal level.
>
> 2. Second, we specifically design our distillation mechanism to address the unique plasticity-stability challenge in neural resets - preserving crucial knowledge while maintaining adaptability. The introduction of bi-directional KL divergence loss, tailored for this intra-network knowledge transfer, enables precise control over knowledge preservation and adaptation at the neuron level. This specialized application and adaptation of distillation principles creates a fundamentally new approach to maintaining neural plasticity.
>
> Response to 1.2 and Questions1
>
> While previous works demonstrate quick recovery in some scenarios, our research reveals that this is highly **algorithm-dependent**. Specifically, algorithms with limited replay buffers like PPO face significant recovery challenges compared to SAC's quick recovery with large buffers. Moreover, our approach addresses a broader goal beyond simple recovery: maintaining continuous learning efficiency throughout training.
>
> Regarding computational costs, our method introduces modest overhead through additional distillation epochs (3%~35% as shown in Table 2). We have added a detailed computational cost comparison between our distillation approach and direct recovery training in Appendix 4.5. While our results indicate that the distillation process consumes more computational time due to our conservative setting of the distillation loss coefficient (0.01), this parameter can be optimized to reduce the computational overhead.
>
> More importantly, Fig. 7 demonstrates that this computational investment yields substantial performance improvements. In contrast, simply adding extra training steps after reset only achieves marginally better results than cycle reset, and generally underperforms compared to basic PPO. This suggests that within the PPO framework, merely increasing training steps is insufficient to fully recover the lost knowledge, even when achieving similar performance metrics.
>
> Response to 3.1
>
> While [2] correctly identifies that critics experience more severe plasticity loss, we argue that addressing plasticity in PPO remains both crucial and insightful. Unlike value-based methods with large replay buffers, PPO's on-policy nature and limited buffer size create distinct plasticity challenges due to continuous distribution shifts in training data. Therefore, this makes PPO an important test case for plasticity enhancement.
>
> The observation that PPO+Cycle alone brings no improvement actually strengthens our argument: it demonstrates that plasticity loss in PPO requires a more sophisticated solution than simple weight resetting. This finding motivated our development of the distillation mechanism, which effectively addresses both the plasticity maintenance and knowledge preservation challenges. Our results suggest that PPO performance limitations may be related to plasticity constraints than fundamental algorithmic limitations.
>
> Response to 3.2
>
> We have added experiments with SAC in Appendix A.5 to enable direct comparison. While these new results demonstrate our method's advantages in optimal conditions, we maintain that our original PPO results, even if sub-optimal, provide valuable insights. Our method's ability to achieve superior performance when baselines converge to sub-optimal solutions specifically highlights its effectiveness in overcoming learning barriers. The combination of optimal SAC results and challenging PPO scenarios strengthens our argument - our approach not only excels under favorable conditions but also demonstrates significant improvements when traditional methods struggle.
>
>
>
> Response to Questions 2
>
> Our research primarily addresses the plasticity challenges in on-policy RL methods, where only a small replay buffer (e.g., 8,192 samples in our experiments) can be utilized to recover useful knowledge that might be temporarily lost during plasticity-enhancing resets. In contrast, off-policy methods can leverage much larger replay buffers (typically 1M samples) to mitigate such knowledge loss. Although we focus on on-policy methods due to their inherent limitations in knowledge recovery, we have also included comprehensive experimental results with SAC in Appendix A.5 to provide a more thorough validation of our approach. These supplementary results demonstrate that our method's benefits extend beyond its intended scope, showing improvements even in off-policy settings.

---

> > ### Author Response · Authors · 2024-12-01
> >
> > As the discussion period ends soon, we would appreciate your feedback on our responses to your review comments. We are ready to address any remaining concerns or questions you may have. If our revisions have addressed your concerns, we hope you would consider adjusting the paper's rating. Thank you again for your time and valuable input on our paper.

---

> > > ### Author Response · Authors · 2024-12-03
> > >
> > > Dear Reviewer Tisv,
> > >
> > > I hope this message finds you well. As the discussion period is ending soon, we wanted to reach out regarding our manuscript. We are grateful for your feedback which has been instrumental in improving our work. We would appreciate if you could review our responses and consider adjusting the paper's rating based on our revisions. We believe we have thoroughly addressed the points you raised.
> > >
> > > Thank you for your time and expertise throughout this review process.

---

### Author Response · Authors · 2024-12-04
**Summary of Review Response**

## Motivation
We appreciate the reviewers' thorough examination of our research foundations. Our study addresses fundamental plasticity challenges in neural networks, particularly focusing on maximizing neuron utilization. Through extensive studies, we demonstrated that complete neuron reset, while seemingly aggressive, is necessary for maximizing network potential. This insight led us to develop a balanced approach using distillation as a recovery mechanism.

## Experiment
Our choice of PPO as the primary testing ground was deliberate and strategic. PPO presents unique challenges due to its small replay buffer and sensitivity to data distribution shifts, making it an ideal candidate for validating our approach. We reasoned that if our methods could succeed in this challenging environment, they would likely generalize well to other scenarios. This hypothesis was later confirmed through our expanded experiments.

Following reviewer suggestions, we substantially expanded our experimental framework:
- Extended our studies (Appendix A4.1) by implementing ReLU-based PPO alongside initial tanh-based versions
- Broadened algorithm coverage by incorporating SAC implementations (Appendix A5.1)
- Performed comprehensive ablation studies examining cycle reset proportions (Appendix A4.4) and inner distillation effects (Appendix A4.5)

Our expanded experimental results strongly validate our approach through three main findings:
- Cycle reset experiments demonstrate that resetting all neurons effectively maintains small weight norms while preserving large gradient flow (Figures 19-20)
- Inner distillation experiments show superior knowledge preservation compared to recovery training, suggesting improved sample efficiency with increased distillation (Figure 21)
- Implementation tests across P3O and SAC+SBP confirm our method's flexibility, demonstrating effectiveness in both on-policy and off-policy scenarios (Figure 23)

These experimental results validate both the necessity and effectiveness of each algorithm component while demonstrating the comprehensive capabilities of our approach.

## Enhanced Plasticity Metrics
Our expanded evaluation metrics demonstrate the robustness of our approach:
- Weight norm analysis (Figures 5, 16, 26)
- Gradient norm measurements (Figures 6, 13, 27)
- Dormant ratio tracking (Figures 12, 24)
- Covariance analysis (Figures 28-30)

The results show consistently desirable characteristics across different settings: small weight norms, low dormant ratio, and large gradient values. The covariance heatmap analysis further validates P3O's superior capability in maintaining both enhanced plasticity and effective neuron utilization.

## Concept
We have carefully refined our algorithm naming to Sustainable Backup Propagation(SBP) and refined the neuron regeneration concept. Neuron regeneration provides a more intuitive and specific description compared to the abstract plasticity-stability trade-off term. The biological analogy of regeneration serves as an intuitive framework for understanding complex network intervention. This precise conceptualization enhances clarity in the field while facilitating meaningful discussions about neuron plasticity maintenance.

## Conclusion
Our comprehensive experiments demonstrate that SBP achieves effective plasticity maintenance through principled neuron regeneration, providing both valuable insights into neural plasticity and practical solutions. This approach shows consistent improvements across different algorithms, validating its broad applicability. These findings not only address the reviewers' concerns but also open up new possibilities for advancing neural network training methodologies.

---

### Meta-Review · Area_Chair_kBiC · 2024-12-23

**Metareview:**

Reviewers shared various positive comments about the submission, complimenting the positive framing of the approach, the good presentation of the work, and its comprehensive experiments, as well as commenting that this is an interesting direction that warrants further investigation. Through the rebuttal, the authors also improved some clarity issues while also introducing new plasticity metrics that reviewers argued for.

While I would have liked to see stronger participation of the reviewers in the discussion (which may have led to a score increase) and could personally envision this submission (with modifications) being published at ICLR, I'm afraid taking all comments together, the average rating (including some discounting of non-responsive reviewers) does not lean strongly enough towards acceptance for this to be justifiable at this stage. Nevertheless, I would like to strongly encourage the authors to re-submit a modified version and feel optimistic about its chances of acceptance in the future.

**Additional Comments On Reviewer Discussion:**

This submission saw a disappointing reviewer discussion with multiple reminders (apart from reviewer EKtA) not engaging in a discussion despite multiple reminders. Unfortunately, these non-responsive reviewers were also more critical of the submission. Despite this, the authors included several of the suggestions. As a result, I have somewhat discounted the influence of these reviews towards the final decision, although it did change the final outcome.

On the positive hand, reviewer EKtA engaged in a discussion with the authors, resulting in an increase in their soundness and confidence score.

---

### Decision · Program_Chairs · 2025-01-22

Reject